# Conformity Score Averaging for Classification

**Rui Luo** [1]   **Zhixin Zhou** [2]

## Abstract

Conformal prediction provides a robust framework for generating prediction sets with finite-sample coverage guarantees, independent of the underlying data distribution. However, existing methods typically rely on a single conformity score function, which can limit the efficiency and informativeness of the prediction sets. In this paper, we present a novel approach that enhances conformal prediction for multi-class classification by optimally averaging multiple conformity score functions. Our method involves assigning weights to different score functions and employing various data splitting strategies. Additionally, our approach bridges concepts from conformal prediction and model averaging, offering a more flexible and efficient tool for uncertainty quantification in classification tasks. We provide a comprehensive theoretical analysis grounded in Vapnik–Chervonenkis (VC) theory, establishing finite-sample coverage guarantees and demonstrating the efficiency of our method. Empirical evaluations on benchmark datasets show that our weighted averaging approach consistently outperforms single-score methods by producing smaller prediction sets without sacrificing coverage. Our code is available at `https://github.com/luo-lorry/Weighting`.

## 1. Introduction

Conformal prediction (Vovk et al., 2005; Manokhin, 2022) is a robust framework that generates prediction sets with finite-sample coverage guarantees, irrespective of the underlying data distribution (Angelopoulos et al., 2023). The fundamental principle of conformal prediction is to construct a prediction set for a new test instance based on the

[1]Department of System Engineering, City University of Hong Kong, China [2]Alpha Benito Research, Los Angeles, USA. Correspondence to: Rui Luo <ruiluo@cityu.edu.hk>, Zhixin Zhou <zzhou@alphabenito.com>.

*Proceedings of the $42^{nd}$ International Conference on Machine Learning*, Vancouver, Canada. PMLR 267, 2025. Copyright 2025 by the author(s).

training data, ensuring that the true label is included with a probability of at least $1 - \alpha$. This coverage assurance holds regardless of the specific point prediction algorithm employed, making conformal prediction a versatile tool for uncertainty quantification in machine learning. In the split conformal prediction framework (Papadopoulos et al., 2002; Lei & Wasserman, 2014; Vovk et al., 2018), the training data is partitioned into a training set and a calibration set. The predictive model is trained on the training subset, while the score functions are evaluated on the calibration subset. The conformal prediction set then comprises all labels whose conformity scores exceed a specific quantile, with the quantile determined by the coverage level.

The choice of score function is critical in determining the efficiency of the resulting prediction sets, especially for multi-class classification. Well-chosen score functions can lead to more informative and precise predictions. This flexibility allows conformal prediction to adapt to the specific characteristics of the data and distribution. Consequently, developing score functions that optimize informativeness and efficiency for various problem settings, including regression (Papadopoulos et al., 2008; 2011; Romano et al., 2019; Kivaranovic et al., 2020; Guan, 2023; Colombo, 2023; 2024) and multi-class classification (Sadinle et al., 2019; Romano et al., 2020; Angelopoulos et al., 2021; Huang et al., 2024; Luo & Zhou, 2024; 2025c), remains an active area of research. This work focuses on enhancing the efficiency of prediction sets for classification tasks. While the underlying idea can extend to regression, this paper will focus on the tasks of multi-class classification. We will also extend the method and theory to basic conformal regression problem.

Our approach assumes the availability of multiple score functions for the same classification task, each differing due to variations in the classification algorithm or the definition of the score. We propose assigning optimal weights to aggregate these score functions. Using a validation set, we determine a threshold to achieve the desired coverage and identify the weight combination that minimizes the prediction set size. The final prediction is then based on this weighted score function. Our aim is to find the optimal weights for linear combinations of score functions, thereby fully leveraging the strengths of existing score functions. While our approach shares similarities with (Yang & Kuchibhotla, 2024), it stands out in three key aspects:

1. **Weighted Averaging of Score Functions:** Instead of selecting the single best-performing score function, our approach combines multiple score functions through optimal weighting. This averaging can yield more efficient prediction sets than any individual score function while maintaining the desired coverage guarantees.

2. **Novel Data Splitting Strategies:** We explore and categorize several data splitting methods to determine the optimal weights for combining score functions. In addition to Validity First Conformal Prediction (VFCP) and Efficiency First Conformal Prediction (EFCP) discussed in (Yang & Kuchibhotla, 2024), we introduce Data Leakage Conformal Prediction (DLCP) and its variant DLCP+, which utilize all available data to enhance weight determination.

3. **Theoretical Foundations Using VC Theory:** We provide a theoretical analysis of our method that leverages Vapnik–Chervonenkis theory to establish coverage guarantees and expected prediction set sizes. This solid mathematical foundation underscores the validity and efficiency of our approach across different data splitting strategies.

Beyond its novel contributions to the conformal prediction literature, our method is closely related to model averaging (Claeskens & Hjort, 2008), a well-established technique in machine learning. Unlike traditional model averaging, which assigns weights to different models to improve prediction accuracy, our method assigns weights to score functions. This distinction requires the development of specific data splitting techniques to ensure the desired coverage guarantees. Consequently, our work can be viewed as an innovative adaptation of model averaging principles to the conformal prediction framework.

The remainder of the paper is organized as follows. In Section 3, we detail our weighted averaging approach and the various data splitting strategies employed. Section 3 presents the theoretical analysis, establishing coverage guarantees and expected prediction set sizes. In Section 4, we demonstrate the effectiveness of our method through experiments. Related works are discussed in Section 5. We conclude in Section 6 and outline future research directions.

**Notations.** $[K]$ denotes the set $\{1, \ldots, K\}$ for positive integer $K$. $\mathcal{I}_1 \sqcup \mathcal{I}_2$ denotes the union of the disjoint sets $\mathcal{I}_1$ and $\mathcal{I}_2$. $\langle a, b \rangle$ denotes the inner product of vector $a$ and $b$. $|C|$ denotes the set size of a finite set $C$.

## 2. Methodology

### 2.1. Conformal Prediction for Classification

We start by assuming that a $K$-class classification algorithm provides $\widehat{p}_y(x)$, which approximates $P(Y = y | X = x)$

---

**Algorithm 1** Split Conformal Prediction

---

**input** Labeled data $\{(x_i, y_i) : i \in \mathcal{I}_{\text{train}} \sqcup \mathcal{I}_{\text{cal}}\}$,
      unlabeled data $\{x_i : i \in \mathcal{I}_{\text{test}}\}$,
      significance level $\alpha$
**output** Prediction set $\widehat{C}(x_i)$ for $i \in \mathcal{I}_{\text{test}}$
1: Train a model $\widehat{p}(x)$ on $\{(x_i, y_i)\}_{i \in \mathcal{I}_{\text{train}}}$.
2: $q_{1-\alpha} \leftarrow \lceil (1 + |\mathcal{I}_{\text{cal}}|)(1 - \alpha) \rceil$-th largest score $s(x_i, y_i)$ for $i \in \mathcal{I}_{\text{cal}}$.
3: **for** $i \in \mathcal{I}_{\text{test}}$ **do**
4:    $\widehat{C}(x_i, q_{1-\alpha}) \leftarrow \{y \in [K] : s(x_i, y) \geq q_{1-\alpha}\}$
5: **end for**

---

for $y \in [K]$. While our method and theoretical analysis do not depend on the accuracy of this approximation, it is beneficial to assume that higher values of $\widehat{p}_y(x)$ indicate a greater likelihood of sample $x$ having label $y$. We consider this training procedure to be performed on a separate dataset, ensuring that $\widehat{p}_y(x)$ is independent of the dataset used in this paper.

We begin by defining a *conformity score function*, $s(x, y)$, which quantifies the agreement between a sample $x$ and a potential label $y$. Higher values of $s(x, y)$ indicate a stronger belief that $y$ is the correct label for $x$. A common choice for this score is the estimated class probability, $s(x, y) = \widehat{p}_y(x)$ (this approach is related to scores used in methods like (Sadinle et al., 2019), though original definitions may vary, e.g., using non-conformity scores).

The conformal prediction procedure for classification can then be outlined. Initially, a predictive model $\widehat{p}_y(x)$ is trained on a dedicated training set $\mathcal{I}_{\text{train}}$. Subsequently, a threshold $q_{1-\alpha}$ is determined using a separate labeled calibration set $\mathcal{I}_{\text{cal}}$. This threshold is chosen such that the conformity scores of true labels, $s(x_i, y_i)$ for $(x_i, y_i) \in \mathcal{I}_{\text{cal}}$, satisfy $s(x_i, y_i) \geq q_{1-\alpha}$ for at least a $1 - \alpha$ proportion of the calibration samples. Finally, for any new test sample $x_j$, this threshold $q_{1-\alpha}$ is used to construct the prediction set as the collection of labels whose conformity scores meet or exceed it: $\widehat{C}(x_j) = \{y \mid s(x_j, y) \geq q_{1-\alpha}\}$. This set represents the upper level set of the scoring function $s(x_j, \cdot)$.

This algorithm constructs prediction sets $\widehat{C}(x_i, q_{1-\alpha})$ for each $i \in \mathcal{I}_{\text{test}}$, based on the conformity scores and a threshold determined by the desired coverage probability $1 - \alpha$. In conformal prediction, we assume the samples in $\mathcal{I}_{\text{cal}}$ and $\mathcal{I}_{\text{test}}$ are exchangeable. The threshold $q_{1-\alpha}$ is chosen to ensure the desired coverage probability under the exchangeability assumption.

### 2.2. Various Score Functions for Classifications

In conformal prediction, the choice of score function $s(x, y)$ critically influences the prediction sets. These functions act

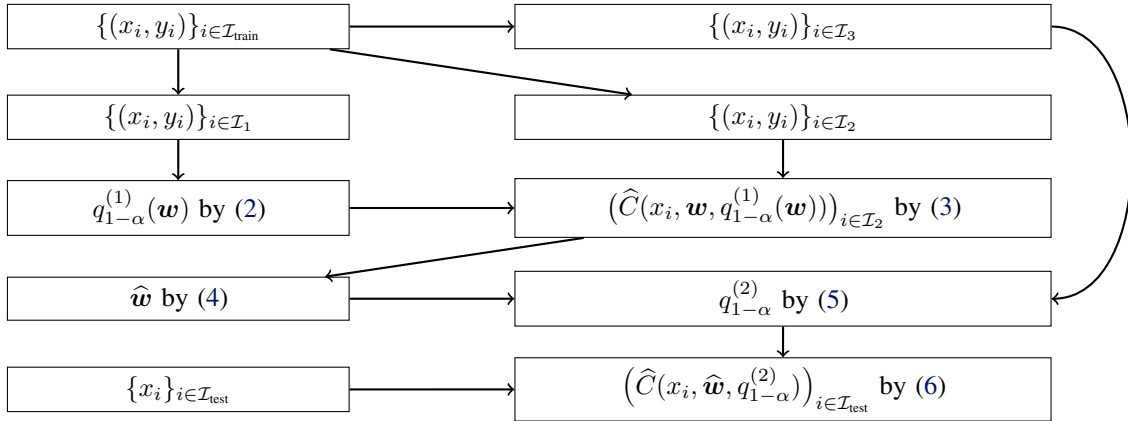

Figure 1. This example illustrates a framework for data splitting into $\mathcal{I}_1, \mathcal{I}_2, \mathcal{I}_3$, and $\mathcal{I}_{\text{test}}$. Algorithm 2 presents the complete procedure. Briefly, $\mathcal{I}_1$ and $\mathcal{I}_2$ are used in Steps 1-2 to select the optimal weight $\widehat{\mathbf{w}}$, while $\mathcal{I}_3$ is used in Step 3 as the calibration set for $\mathcal{I}_{\text{test}}$ predictions. We present four options: VFCP, EFCP, DLCP, and DLCP+. Their coverage and size properties are discussed theoretically in Section 3 and empirically in Section 4.

as conformity measures that quantify how appropriately a label matches an input.

Score functions are determined by two primary factors: (1) The predictive model's quality in estimating $\widehat{p}_y(x) \approx \mathbb{P}(Y = y \mid X = x)$, and (2) The specific methodology for converting these probability estimates into scores. We will consider approaches:

1. Threshold (THR) (Sadinle et al., 2019): $s_{\text{THR}}(x, y) = \widehat{p}_y(x)$

2. Adaptive Prediction Sets (APS) (Romano et al., 2020): $s_{\text{APS}}(x, y) = \sum_{y' \in [K]} \widehat{p}_{y'}(x) \mathbf{1}\{\widehat{p}_{y'}(x) \leq \widehat{p}_y(x)\}$

3. Rank-based (RANK) (Luo & Zhou, 2024): $s_{\text{RANK}}(x, y) = |\{y' \in [K] : \widehat{p}_{y'}(x) < \widehat{p}_y(x)\}|$

These score functions integrate directly with Algorithm 1. While the original formulations include additional refinements to achieve exact $1 - \alpha$ coverage guarantees, these refinements do not materially affect our combination methodology and are omitted for simplicity. Our theoretical analysis remains invariant to specific score function design choices. Other notable approaches like RAPS (Angelopoulos et al., 2021) and SAPS (Huang et al., 2024) demonstrate alternative methodologies for deriving scores from $\widehat{p}_y(x)$.

### 2.3. Averaging Score Functions

In (Yang & Kuchibhotla, 2024), the authors propose their method with multiple score functions. Their approach selects the score function that yields the smallest average prediction set size among the available score functions. They also introduce two data splitting methods: Efficiency First Conformal Prediction (EFCP) and Validity First Conformal Prediction (VFCP).

Given a vector of score functions $s(x, y) = (s_1(x, y), \ldots, s_d(x, y))^\top$, instead of simply choosing the score that provides the smallest prediction set on a calibration set, we propose assigning weights $\mathbf{w} \in \mathcal{W} \subseteq \mathbb{R}^d$ to the conformity scores and defining the weighted score function:

$$\langle \mathbf{w}, s(x, y) \rangle = \sum_{j=1}^{d} \mathbf{w}_j s_j(x, y). \tag{1}$$

By leveraging different scores to create a completely new score function, we expect the weighted score to outperform any individual score. This approach allows for more flexibility in combining the strengths of various score functions, potentially leading to improved prediction set efficiency.

### 2.4. The Optimal Weight and the Threshold

In this section, we present a detailed procedure for determining the optimal weight vector $\mathbf{w}$ in the context of the weighted score approach introduced in (1). This procedure extends Algorithm 1 by incorporating an additional step to minimize the expected size of the prediction set. The following steps outline the process:

1. Train the model $\widehat{p}$ using samples in $\mathcal{I}_{\text{train}}$ and obtain the score functions $s_1, \ldots, s_d$.

2. Extract $\mathcal{I}_1, \mathcal{I}_2, \mathcal{I}_3$ from the $\mathcal{I}_{\text{val}}$. We can first assume they are partitions of $\mathcal{I}_{\text{val}}$. More options will be described in Section 2.5.

3. Let $\mathcal{W} \subseteq \mathbb{R}^d$ be the set for the candidates of $\mathbf{w}$ candidates. A typical choice is the a discretized simplex $\mathcal{W} = \Delta^{d-1}$. We systematically explore candidate

**Algorithm 2** Conformal Score Averaging

**input** Labeled data $\{(x_i, y_i) : i \in \mathcal{I}_{\text{val}}\}$,
       unlabeled data$\{x_i : i \in \mathcal{I}_{\text{test}}\}$,
       significance level $\alpha$,
       score functions $s = (s_1, \ldots, s_d)$,
       set of possible weights: $\mathcal{W} \subseteq \mathbb{R}^d$
**output** Prediction set $\widehat{C}(x_i)$ for $i \in \mathcal{I}_{\text{test}}$
1: Decide $\mathcal{I}_1 \subseteq \mathcal{I}_{\text{val}}$, $\mathcal{I}_2 \subseteq \mathcal{I}_{\text{val}} \cup \mathcal{I}_{\text{test}}$, and $\mathcal{I}_3 \subseteq \mathcal{I}_{\text{val}}$.
2: **for** $w \in \mathcal{W}$ **do**
3:    ▷ Compute quantile for every $w$:
4:    $q_{1-\alpha}^{(1)}(w) \leftarrow \text{Calibration}(\mathcal{I}_1, s, w, \alpha)$
5:    **for** $i \in \mathcal{I}_2$ **do**
6:       ▷ Find a temporary prediction set:
7:       $\widetilde{C}(x_i; w) \leftarrow \text{Evaluation}(x_i, s, w, q_{1-\alpha}^{(1)}(w))$
8:       ▷ Compute average prediction set size:
9:       $S(w) = \frac{1}{|\mathcal{I}_2|} \sum_{i \in \mathcal{I}_2} |\widetilde{C}(x_i; w)|$
10:   **end for**
11: **end for**
12: ▷ Find the most efficient $w$:
13: $\widehat{w} \leftarrow \arg\min_{w \in \mathcal{W}} S(w)$
14: ▷ Compute quantile for $\widehat{w}$:
15: $q_{1-\alpha}^{(2)} \leftarrow \text{Calibration}(\widehat{w}, \mathcal{I}_3, \alpha)$
16: ▷ Find the final prediction set:
17: **for** $i \in \mathcal{I}_{\text{test}}$ **do**
18:   $\widehat{C}(x_i, \widehat{w}) \leftarrow \text{Evaluation}(x_i, s, \widehat{w}, q_{1-\alpha}^{(2)})$
19: **end for**
20: **Return** $\widehat{C}(x_i, \widehat{w})$ for $i \in \mathcal{I}_{\text{test}}$

---

**Function 1** Calibration

**input** Labeled data $\{(x_i, y_i) : i \in \mathcal{I}\}$,
       score functions $s = (s_1, \ldots, s_j)$,
       weight $w \in \mathcal{W}$,
       significance level $\alpha$
**output** $q_{1-\alpha}$
1: $q_{1-\alpha} \leftarrow \lceil (1 + |\mathcal{I}|)(1 - \alpha) \rceil$-th largest score $\langle w, s(x_i, y_i) \rangle$ from scores computed for $i \in \mathcal{I}$.
2: **Return** $q_{1-\alpha}$.

---

**Function 2** Evaluation

**input** Unlabeled data $x$,
       score functions $s = (s_1, \ldots, s_j)$,
       weight $w \in \mathcal{W}$,
       quantile $q$
**output** $\widehat{C}(x)$
1: $\widehat{C}(x) \leftarrow \{y \in [K] : \langle w, s(x, y) \rangle \geq q_{1-\alpha}\}$.
2: **Return** $\widehat{C}(x)$

---

7. For given $\widehat{w}$, $\mathcal{I}_3$ is treated as the calibration set Algorithm 1, and we obtain:

$$q_{1-\alpha}^{(2)} = \lceil (1 + |\mathcal{I}_3|)(1 - \alpha) \rceil\text{-th largest} \quad (5)$$
$$\text{value of} \quad \{\langle \widehat{w}, s(x_i, y_i) \rangle : i \in \mathcal{I}_3\}.$$

If $\mathcal{I}_1 = \mathcal{I}_3$, then this quantile has been computed in (2), i.e., $q_{1-\alpha}^{(2)} = q_{1-\alpha}^{(1)}(\widehat{w})$.

8. The final output is the confidence set for samples $i \in \mathcal{I}_{\text{test}}$ in the test set:

$$\widehat{C}(x_i, \widehat{w}) = \{y \in [K] : \langle \widehat{w}, s(x_i, y) \rangle \geq q_{1-\alpha}^{(2)}\}. \quad (6)$$

This procedure comprises two primary steps: **Threshold Calibration**: Determining appropriate thresholds using Equations (2) and (5) based on the calibration subsets $\mathcal{I}_1$ and $\mathcal{I}_3$. **Prediction Set Evaluation**: Constructing the prediction sets for $\mathcal{I}_2$ and $\mathcal{I}_{\text{test}}$ as outlined in Equations (3) and (6). These steps are encapsulated in Function 1 and Function 2, respectively. The entire procedure of the proposed method is summarized in Algorithm 2.

### 2.5. Data Splitting

In Algorithm 2, the method for splitting the data into $\mathcal{I}_1$, $\mathcal{I}_2$, and $\mathcal{I}_3$ has not been specified. To explore potential data splitting approaches, we first highlight the following two key observations. Firstly, After determining $\widehat{w}$, the calibration procedure (5) and prediction set construction (6) mirror the step of finding quantile and prediction set in Algorithm 1. Prior steps aim to identify a weight vector $w$ optimizing algorithm performance. Secondly, (2) and (5) find quantiles, requiring sample labels. Equations (3) and (6)

---

weights via grid search with step size $\varepsilon = 0.01$, as detailed in Appendix A.

4. Calculate the threshold for every $w \in \mathcal{W}$ on $\mathcal{I}_1$. In the set $\mathcal{I}_1$, calculate the threshold for every $w \in \mathcal{W}$:

$$q_{1-\alpha}^{(1)}(w) = \lceil (1 + |\mathcal{I}_1|)(1 - \alpha) \rceil\text{-th largest} \quad (2)$$
$$\text{value of} \quad \{\langle w, s(x_i, y_i) \rangle : i \in \mathcal{I}_1\}.$$

5. Determine the prediction set for data in $\mathcal{I}_2$. For a given $w$ and threshold $q_{1-\alpha}^{(1)}(w)$, define the prediction set for each sample $x_i, i \in \mathcal{I}_2$ as:

$$\widetilde{C}(x_i, w) = \{y \in [K] : \langle w, s(x_i, y) \rangle \geq q_{1-\alpha}^{(1)}(w)\}. \quad (3)$$

These are the prediction sets for all $w \in \mathcal{W}$. Our goal is to minimize the prediction set size. This intuitively leads to the next step.

6. The optimal weight vector $\widehat{w}$ is obtained by minimizing the empirical prediction set size:

$$\widehat{w} \in \arg\min_{w \in \mathcal{W}} \frac{1}{\mathcal{I}_2} \sum_{i \in \mathcal{I}_2} |\widetilde{C}(x_i, w)|. \quad (4)$$

define prediction sets, needing only feature $x$. Thus, the test set can be included in (3) and (6).

Based on the observations above, we introduce the following four possible ways of data splitting.

(a) Validity First Conformal Prediction (VFCP) (Yang & Kuchibhotla, 2024): $\mathcal{I}_1 = \mathcal{I}_2 \subseteq \mathcal{I}_{\text{val}}$, and $\mathcal{I}_3 = \mathcal{I}_{\text{val}} \setminus \mathcal{I}_1$. Divides $\mathcal{I}_{\text{train}}$ into two partitions. Marginal coverage probability is guaranteed to be at least $1 - \alpha$ under exchangeability of samples in $\mathcal{I}_3$ and $\mathcal{I}_{\text{test}}$.

(b) Efficiency First Conformal Prediction (EFCP) (Yang & Kuchibhotla, 2024): $\mathcal{I}_1 = \mathcal{I}_2 = \mathcal{I}_3 = \mathcal{I}_{\text{train}}$. Uses all training data to determine $\widehat{w}$, resulting in more accurate estimation of optimal $w$. This method equires stronger assumptions for coverage guarantee.

(c) Data Leakage Conformal Prediction (DLCP): $\mathcal{I}_1 = \mathcal{I}_3 = \mathcal{I}_{\text{train}}$, and $\mathcal{I}_2 = \mathcal{I}_{\text{test}}$. Minimizes prediction set on $\mathcal{I}_{\text{test}}$ in finding $\widehat{w}$ in (4). Called "data leakage" as it uses test data in training procedure of $\widehat{w}$.

(d) Data Leakage Conformal Prediction+ (DLCP+): $\mathcal{I}_1 = \mathcal{I}_3 = \mathcal{I}_{\text{train}}$ and $\mathcal{I}_2 = \mathcal{I}_{\text{train}} \cup \mathcal{I}_{\text{test}}$. Uses all available data in each step, including $\mathcal{I}_{\text{test}}$ in $\mathcal{I}_2$ to maximize sample size for finding $\widehat{w}$ in (4).

There are various possible methods for data splitting. In the following sections, we focus on these four specific approaches, examining their theoretical properties and evaluating their performance through experiments.

## 3. Theoretical Analysis

### 3.1. Overview of the Results

We investigate the theoretical properties of the proposed methods in terms of validity and efficiency.

**Validity:** We assess whether the output prediction set achieves the desired coverage rate of $1 - \alpha$. For the VFCP method, the coverage rate is guaranteed under the exchangeability assumption. However, for the other three methods, establishing validity is more complex due to the selection bias introduced by $\widehat{w}$. Consequently, our aim is to demonstrate that the optimization over $w \in \mathcal{W}$ has small impact on validity.

**Efficiency:** We observe that the prediction set attains the smallest possible expected size only if the true conditional probabilities $p(y|x)$ are known. In this section, let us define

$$\widehat{C}(x, \boldsymbol{w}, q) = \{y \in [K] : \langle \boldsymbol{w}, s(x, y) \rangle \geq q\}.$$

We can define the population level optimal weight

$$\boldsymbol{w}^* = \arg\min_{\boldsymbol{w} \in \mathcal{W}} \min_{q \in \mathbb{R}} \mathbb{E}\left[|\widehat{C}(X, \boldsymbol{w}, q)|\right]$$
$$\text{s.t.} \quad \mathbb{P}\left(Y \in \widehat{C}(X, \boldsymbol{w}, q)\right) \geq 1 - \alpha.$$

$\boldsymbol{w}^*$ is the weight vector that minimizes the expected prediction set size while ensuring that, together with an appropriate threshold $q$, the prediction set maintains a desired coverage rate of $1 - \alpha$. Equivalently, we can define

$$q_{1-\alpha}(\boldsymbol{w}) := \sup\{q \in \mathbb{R} : \mathbb{P}(Y \in \widehat{C}(X, \boldsymbol{w}, q)) \geq 1 - \alpha\}$$

and using this definition, we define

$$\boldsymbol{w}^* \in \arg\min_{\boldsymbol{w} \in \mathcal{W}} \mathbb{E}\left[|\widehat{C}(X, \boldsymbol{w}, q_{1-\alpha}(\boldsymbol{w}))|\right].$$

Since $\boldsymbol{w}^*$ and $q_{1-\alpha}(\boldsymbol{w}^*)$ are deterministic once $\alpha$ and the true distribution are given, we can succinctly denote the optimal prediction set for $x$:

$$\widehat{C}_{1-\alpha}^*(x) := \widehat{C}(x, \boldsymbol{w}^*, q_{1-\alpha}(\boldsymbol{w}^*)).$$

Our analysis of efficiency focuses on the difference between the size of the prediction set produced by the proposed methods using $\widehat{w}$ and the expected size of $\widehat{C}_{1-\alpha}^*(x)$. In other words, we analyze the discrepancy between the prediction set sizes generated by $\widehat{w}$ and the optimal $\boldsymbol{w}^*$. We will demonstrate that this difference is negligible given a sufficiently large dataset.

We note that this analysis differs from verifying whether $\widehat{w}$ converges to $\boldsymbol{w}^*$, which would require assumptions about the smoothness of the function mapping the weight to the prediction set size. Instead, our analysis makes fewer assumptions on the optimal prediction set size.

### 3.2. Results by Vapnik–Chervonenkis Theory

We begin by defining the following events on the probability space of pairs of exchangeable random variables $(X_i, Y_i)_{i \in \mathcal{I}}$. The event $\Omega(\mathcal{I}, \eta)$ is defined as:

$$\sup_{\boldsymbol{w} \in \mathbb{R}^d, q \in \mathbb{R}} \left| \frac{1}{|\mathcal{I}|} \sum_{i \in \mathcal{I}} \mathbf{1}\{\langle \boldsymbol{w}, s(X_i, Y_i) \rangle \geq q\} \right.$$
$$\left. - \mathbb{E}_{X,Y}[\mathbf{1}\{\langle \boldsymbol{w}, s(X, Y) \rangle \geq q\}] \right| \leq \eta, \quad (7)$$

where $(X, Y)$ has the same distribution as $(X_1, Y_1)$ and the expectation is for the joint distribution of $(X, Y)$. Let $\Gamma(\mathcal{I}, \xi)$ denotes the event

$$\sup_{\boldsymbol{w} \in \mathbb{R}^d, q \in \mathbb{R}} \left| \frac{1}{|\mathcal{I}|} \sum_{i \in \mathcal{I}} \sum_{y \in [K]} \mathbf{1}\{\langle \boldsymbol{w}, s(X_i, y) \rangle \geq q\} \right.$$
$$\left. - \mathbb{E}_X \left[ \sum_{y \in [K]} \mathbf{1}\{\langle \boldsymbol{w}, s(X, y) \rangle \geq q\} \right] \right| \leq \xi, \quad (8)$$

Here, the expectation is taken over $X$ only.

$\Omega(\mathcal{I}, \eta)$ is about the uniform concentration of the coverage, and $\Gamma(\mathcal{I}, \xi)$ is about the uniform concentration of the prediction set size. This is because by the definition of $\widehat{C}_{1-\alpha}$

in (2) or (5),

$$\mathbf{1}\{\langle \boldsymbol{w}, s(x,y)\rangle \geq q\} = \mathbf{1}\{y \in \widehat{C}_{1-\alpha}(x, \boldsymbol{w}, q)\} \quad \text{and}$$

$$\sum_{y \in [K]} \mathbf{1}\{\langle \boldsymbol{w}, s(x,y)\rangle \geq q\} = \big|\widehat{C}_{1-\alpha}(x, \boldsymbol{w}, q)\big|.$$

The following lemma shows that these two events hold with high probability.

**Lemma 1.** *Suppose the samples in $\mathcal{I}$ are i.i.d., then*

(a) $\Omega\left(\mathcal{I}, 8\sqrt{\frac{(d+1)\log(|\mathcal{I}|+1)}{|\mathcal{I}|}} + \delta\right)$ *hold with probability at least $1 - \exp\left(-\frac{|\mathcal{I}|\delta^2}{2}\right)$.*

(b) $\Gamma\left(\mathcal{I}, 8K\sqrt{\frac{(d+1)\log(|\mathcal{I}|+1)}{|\mathcal{I}|}} + K\delta\right)$ *hold with probability at least $1 - \exp\left(-\frac{|\mathcal{I}|\delta^2}{2}\right)$.*

The proof of this lemma appears in Section B in the appendix. This lemma establishes that $\Omega(\mathcal{I}, \eta)$ and $\Gamma(\mathcal{I}, \xi)$ hold with high probability provided that $\eta, \xi \gtrsim \sqrt{\frac{d\log|\mathcal{I}|}{|\mathcal{I}|}}$. The proof, presented in the appendix, applies subgraph classes from Vapnik-Chervonenkis (VC) theory. Additionally, if the inequalities in Equations (7) and (8) hold for $\boldsymbol{w} \in \mathbb{R}^d$, they naturally extend to $\boldsymbol{w} \in \mathcal{W} \subset \mathbb{R}^d$.

**Remark.** *Part (b) of Lemma 1 limits the generalization of our current theoretical results to classification tasks. Compared to the bound in Equation (7) (for coverage), the bound in Equation (8) (for prediction set size) involves a summation over the $K$ classes. In our proof, this summation is handled using a union bound over $y \in [K]$ when analyzing the concentration of individual terms $\mathbf{1}\{\langle \mathbf{w}, s(X,y)\rangle \geq q\}$. However, for regression tasks where prediction sets are typically intervals and their size is measured by Lebesgue measure, a simple union bound over discrete classes is not directly applicable. Therefore, our primary theoretical analysis focuses on classification.*

*Nonetheless, our framework shows promise for certain regression settings. Consider the class of prediction sets $\mathcal{A} := \{\{y : \langle \mathbf{w}, s(x,y)\rangle \geq t\} : \mathbf{w} \in \mathbb{R}^d, t \in \mathbb{R}\}$. If the individual score functions $s_j(x,y)$ are concave in $y$ (for non-negative weights $w_j \geq 0$), then the weighted score $\langle \mathbf{w}, s(x,y)\rangle$ is also concave in $y$. In such cases, the superlevel sets $\{y : \langle \mathbf{w}, s(x,y)\rangle \geq t\}$ are intervals (or empty, or the whole real line). The class of all intervals in $\mathbb{R}$ has a VC dimension of 2. If the weighted score functions in a regression context produce such interval-valued prediction sets, the analysis of prediction set size concentration might simplify considerably, potentially avoiding the $K$-dependency. This suggests that our method could be extended to handle weighted averages of concave score functions in regression, such as those used in Conformalized Quantile Regression (CQR) (Romano et al., 2019), a direction we leave for future research.*

### 3.3. Consistency of VFCP

The statistical guarantees of VFCP requires a few additional assumptions, e.g., the continuity between the quantile and the prediction set. To get rid off such assumptions and present the result in a neater way, we will only prove a modified version of VFCP. We will change $\alpha$ in (2) to a slightly smaller $\alpha'$. The condition for $\alpha'$ will be specified in the theorem.

**Theorem 1.** *Let $\widehat{C}_{1-\alpha}^{VFCP}(x)$ be the output of Algorithm 2 with the setting of VFCP, and modified as mentioned above. Suppose the samples in $\{(X_i, Y_i)\}_{i \in \mathcal{I}_3 \cup \mathcal{I}_{test}}$ are exchangeable, then for $(X,Y)$ in the test set, the coverage probability*

$$\mathbb{P}\left(Y \in \widehat{C}_{1-\alpha}^{VFCP}(X)\right) \geq 1 - \alpha.$$

*Moreover, if $\Omega(\mathcal{I}_1, \eta_1), \Omega(\mathcal{I}_3, \eta_3)$ and $\Gamma(\mathcal{I}_2, \xi_2)$ are satisfied, and $\alpha' + \eta_1 + \eta_3 \leq \alpha$, then for test sample $X$,*

$$\mathbb{E}\left[\big|\widehat{C}_{1-\alpha}^{VFCP}(X)\big|\right] \leq \mathbb{E}\left[\big|\widehat{C}_{1-\alpha_1}^*(X)\big|\right] + 2\xi_2,$$

*where $1 - \alpha_1 = \frac{1}{|\mathcal{I}_3|}\lceil(1 + |\mathcal{I}_3|)(1 - \alpha')\rceil - \eta_3$.*

The proofs of all theorems appear in Section C and Section D and the appendix.

The validity of VFCP requires minimal assumption (exchangeability), while it is less efficient than other method. We will verify this fact in empirical studies.

### 3.4. Consistency of EFCP

In the setting of EFCP, $\mathcal{I}_1 = \mathcal{I}_2 = \mathcal{I}_3 = \mathcal{I}_{val}$. We denote this set by $\mathcal{I}_{val}$.

**Theorem 2.** *Let $\widehat{C}_{1-\alpha}^{EFCP}(x)$ be the output of Algorithm 2 with the setting of EFCP. Suppose the data in $\mathcal{I}_{val}$ and $\mathcal{I}_{test}$ are independent and $\Omega(\mathcal{I}_{val}, \eta_{val})$ is satisfied. Then for $(X,Y)$ in the test set, the coverage probability*

$$\mathbb{P}\left(Y \in \widehat{C}_{1-\alpha}^{EFCP}(X)\right) \geq 1 - \alpha_1 \quad \text{where}$$

$$1 - \alpha_1 = \frac{1}{|\mathcal{I}_{val}|}\lceil(1 + |\mathcal{I}_{val}|)(1 - \alpha)\rceil - \eta_{val}.$$

*Moreover, if $\Gamma(\mathcal{I}_{val}, \xi_{val})$ is satisfied, then for any $X$ in the test set,*

$$\mathbb{E}\left[\big|\widehat{C}_{1-\alpha}^{EFCP}(X)\big|\right] \leq \mathbb{E}\left[\big|\widehat{C}_{1-\alpha_1}^*(X)\big|\right] + 2\xi_{val}.$$

### 3.5. Consistency of DLCP

In the setting of DLCP, $\mathcal{I}_1 = \mathcal{I}_3 = \mathcal{I}_{val}$ and $\mathcal{I}_2 = \mathcal{I}_{test}$. We will denote this by $\mathcal{I}_{val}$ and $\mathcal{I}_{test}$ respectively. In this case, the test set and $\widehat{w}$ become dependent. The prediction set size is presented differently than previous results.

**Theorem 3.** *Let $\widehat{C}_{1-\alpha}^{DLCP}(x))$ be the output of Algorithm 2 for a test sample $x$ with the setting of DLCP. Suppose*

$\Omega(\mathcal{I}_{val}, \eta_{val})$ and $\Omega(\mathcal{I}_{test}, \eta_{test})$ hold, then the coverage proportion satisfies

$$\frac{1}{|\mathcal{I}_{test}|} \sum_{i \in \mathcal{I}_{test}} \mathbf{1}\{y_i \in \widehat{C}_{1-\alpha}^{DLCP}(x_i)\} \geq 1 - \alpha_1 - \eta_{test} \quad \text{where}$$

$$1 - \alpha_1 = \frac{1}{|\mathcal{I}_{val}|} \lceil (1 + |\mathcal{I}_{val}|)(1 - \alpha) \rceil - \eta_{val}.$$

In addition, if $\Gamma(\mathcal{I}_{test}, \xi_{test})$ holds, then

$$\frac{1}{|\mathcal{I}_{test}|} \sum_{i \in \mathcal{I}_{test}} |\widehat{C}_{1-\alpha}^{DLCP}(x_i)| \leq \mathbb{E}\left[ |\widehat{C}_{1-\alpha_1}^{*}(X)| \right] + \xi_{test}.$$

### 3.6. Consistency of DLCP+

Similar as the DLCP setting, $\widehat{\boldsymbol{w}}$ depends on the test set, so the result of prediction set size is presented in a similar way. In the following theorem, we use $\mathcal{I}_2$ to denote $\mathcal{I}_{val} \cup \mathcal{I}_{test}$.

**Theorem 4.** *Let $\widehat{C}_{1-\alpha}^{DLCP+}(x, \widehat{\boldsymbol{w}})$ be the output of Algorithm 2 with the setting of DLCP+. Suppose $\Omega(\mathcal{I}_{val}, \eta_{val})$ and $\Omega(\mathcal{I}_{test}, \eta_{test})$ hold, then the coverage proportion*

$$\frac{1}{|\mathcal{I}_{test}|} \sum_{i \in \mathcal{I}_{test}} \mathbf{1}\{y_i \in \widehat{C}_{1-\alpha}^{DLCP+}(x_i, \widehat{\boldsymbol{w}})\} \geq 1 - \alpha' - \eta_{test}$$

$$\text{where} \quad 1 - \alpha' = \frac{1}{|\mathcal{I}_{val}|} \lceil (1 + |\mathcal{I}_{val}|)(1 - \alpha) \rceil - \eta_{val}.$$

In addition, if $\Gamma(\mathcal{I}_2, \xi_2)$ holds, then for test sample $X$,

$$\mathbb{E}[|\widehat{C}_{1-\alpha}^{DLCP+}(X)|] \leq \mathbb{E}\left[ |\widehat{C}_{1-\alpha'}^{*}(X)| \right] + 2\xi_2.$$

### 3.7. Conclusion of Theoretical Results

We integrate the results of Lemma 1 with our main theorems. The Lemma 1 indicates that as the size of the set $\mathcal{I}$ approaches infinity, the events $\Omega(\mathcal{I}, \eta)$ and $\Gamma(\mathcal{I}, \xi)$ hold with probability $1 - o(1)$ even if $\eta, \xi \to 0$ is at a sufficient slow rate. Consequently, the terms $\eta_i$ and $\xi_i$ in our theorems become negligible. This suggests that the proposed methods achieve coverage rates close to $1 - \alpha$ and exhibit near-optimal efficiency.

## 4. Experiments

### 4.1. Score Weighting

We conducted an experiment to compare the performance of various score functions score functions discussed in Section 2.2: THR (Sadinle et al., 2019), APS (Romano et al., 2020), and RANK (Luo & Zhou, 2024) given by a pretrained classifier $\widehat{p}_y(x)$. Additional details about implementation can be found in Section E in the appendix. Throughout the experiment, we assumed that a pretrained classifier $\widehat{p}_y(x)$ was available, and the split of the dataset unseen during the training of the pretrained classifier.

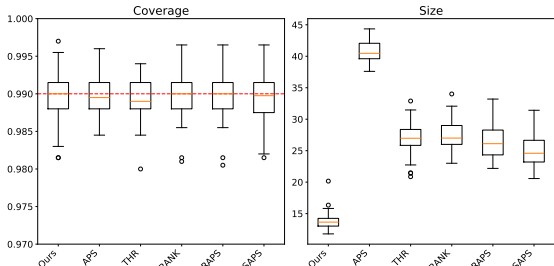

*Figure 2.* Boxplot comparison of different score functions at a significance level of $\alpha = 0.01$ on CIFAR-100. Our weighted combination method achieves the guaranteed coverage of 99% while maintaining the smallest prediction set size.

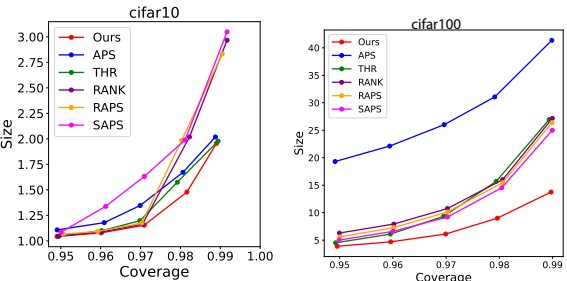

*Figure 3.* Comparison of size vs. coverage for various score functions and our proposed method across $\alpha$ values (0.01-0.05). Our weighted combination method (red) consistently outperforms the other baseline methods by achieving the desired coverage rate with smaller prediction set sizes.

In the experiments on CIFAR-10 and CIFAR-100, testing images, which were not used during the pretraining of the model, were used as the $\mathcal{I}_{train}$ and $\mathcal{I}_{test}$ sets. The experiments were performed for different significance levels $\alpha$ ranging from 0.01 to 0.05. 100 runs with different index splits were conducted to ensure robustness. We have additional experiments on data splitting ratio, which can be found in Section G in the supplementary file.

The primary objective of our experiments was to evaluate the performance of the proposed weighted score function in comparison with three foundational base score functions: APS, THR, and RANK. Additionally, we compared our method against two competitive baseline score functions, RAPS (Angelopoulos et al., 2021) and SAPS (Huang et al., 2024).

Figure 2 compares coverage and prediction set sizes across methods at $\alpha = 0.01$ on CIFAR-100, while Figure 3 evaluates performance across $\alpha \in [0.01, 0.05]$. Using VFCP splits for consistency, our method achieves guaranteed coverage with the smallest prediction sets overall. The advantages are particularly pronounced on CIFAR-10 across all $\alpha$ values and on CIFAR-100 for $\alpha \leq 0.02$.

| | $\alpha = 0.01$ | | $\alpha = 0.05$ | |
|---|---|---|---|---|
| Method | Coverage | Size | Coverage | Size |
| VFCP | 0.990 (0.003) | 13.782 (1.114) | 0.950 (0.005) | 3.890 (0.266) |
| EFCP | 0.989 (0.003) | 13.306 (0.464) | 0.949 (0.005) | 3.754 (0.096) |
| DLCP | 0.989 (0.003) | 13.298 (0.461) | 0.949 (0.005) | 3.752 (0.097) |
| DLCP+ | 0.989 (0.003) | 13.299 (0.459) | 0.949 (0.005) | 3.753 (0.097) |
| APS | 0.990 (0.003) | 40.217 (1.786) | 0.949 (0.006) | 19.545 (1.022) |
| THR | 0.989 (0.003) | 26.949 (2.198) | 0.949 (0.006) | 4.519 (0.381) |
| RANK | 0.990 (0.003) | 27.161 (2.421) | 0.950 (0.006) | 6.298 (0.395) |
| RAPS | 0.990 (0.003) | 26.403 (2.505) | 0.950 (0.005) | 5.581 (0.263) |
| SAPS | 0.990 (0.003) | 25.096 (2.298) | 0.950 (0.006) | 5.036 (0.302) |

*Table 1.* Coverage and size of different methods for $\alpha = 0.01$ and $\alpha = 0.05$ on CIFAR-100 dataset. Results are shown as mean (standard deviation). The first four methods corresponds to the splitting methods in Section 2.5.

Notably, THR exhibits strong performance on less challenging tasks (CIFAR-10), where our weighting scheme naturally assigns it dominant weights. In these cases, score averaging provides limited improvement. However, for complex multi-class scenarios (CIFAR-100) or stringent significance levels ($\alpha < 0.05$), our method demonstrates clear superiority by optimally combining score functions.

Furthermore, to explore the influence of different data splitting strategies (as discussed in Section 2.4), we conducted additional comparisons of our weighted score function using various split approaches. We also included the performance results of the five individual score functions when utilizing the VFCP split method. Table 1 provides a comprehensive summary of the coverage and size metrics at significance levels of $\alpha = 0.01$ and $\alpha = 0.05$. These results highlight the effectiveness of the different data splitting strategies. It is important to note that while APS aims to achieve conditional coverage, the other methods, including our approach, do not specifically target conditional coverage. Consequently, it is not surprising that the alternative methods exhibit better efficiency compared to APS. Additional experimental results comparing our method with baseline approaches and Synergy Conformal Prediction on MNIST, Fashion-MNIST, and ImageNet-Val datasets are provided in Appendix F.

### 4.2. Model Weighting

In addition to choosing the score functions in Section 2.2, we further conducted an experiment to compare the performance of various models and their weighted combinations for prediction set construction on CIFAR-10 and CIFAR-100. For CIFAR-10, we used the models ResNet-56, ShuffleNetV2 (1.0x), and VGG16-BN. For CIFAR-100, we used the models VGG16-BN, RepVGG-A2, and MobileNetV2 (1.0x). The performance of the weighted combination method was compared against individual models.

Specifically, for each of the four score functions: THR, APS, RAPS, and SAPS, we generated prediction sets by combining the scores of different models using weights

selected by our proposed method. The results are visualized in Figures 4 and 5, which demonstrate the effectiveness of our weighted combination method in terms of weighing scores of different models.

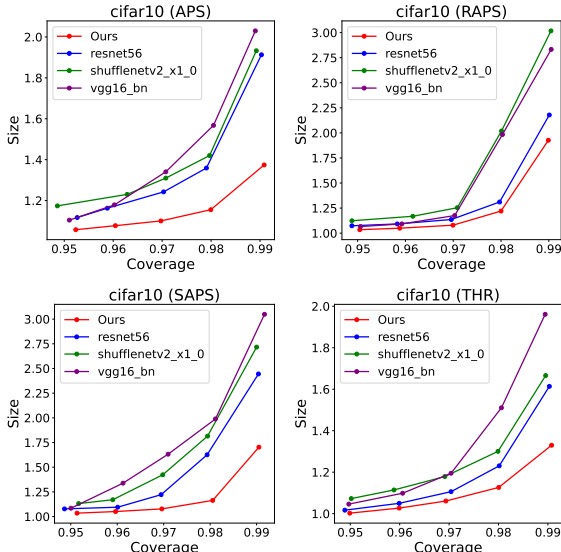

*Figure 4.* Across various score functions, our weighted combination of models outperformed any individual model and achieved optimal size on the CIFAR-10 dataset across $\alpha$ values (0.01–0.05).

## 5. Related Work

Conformal prediction (CP) (Vovk et al., 2005) is a methodology designed to generate prediction regions for variables of interest, facilitating the estimation of model uncertainty by providing prediction sets rather than point estimates. CP has been successfully applied to both classification (Luo & Zhou, 2024; Luo & Colombo, 2024) and regression tasks (Luo & Zhou, 2025c;d). Its flexibility allows adaptation to various real-world scenarios, including segmentation (Luo & Zhou, 2025a), games (Luo et al., 2024; Bao et al., 2025), time-series forecasting (Su et al., 2024), and graph-based applications (Luo et al., 2023; Tang et al., 2025; Luo & Zhou, 2025b; Wang et al., 2025; Luo & Colombo, 2025; Zhang et al., 2025).

Our work builds upon advances in conformal prediction, particularly in model averaging and calibration. In model aggregation, Yang et al. (Yang & Kuchibhotla, 2024) introduced two selection algorithms to minimize the width of prediction intervals by aggregating and selecting from multiple regression estimators. Various VC dimension techniques have been employed in (Yang & Kuchibhotla, 2024) and (Candès et al., 2023): in Section D.2 of (Yang & Kuchibhotla, 2024), the uniform probability bound is established over a finite set, whereas (Candès et al., 2023) considers maximization over real numbers. However, neither approach incorporates

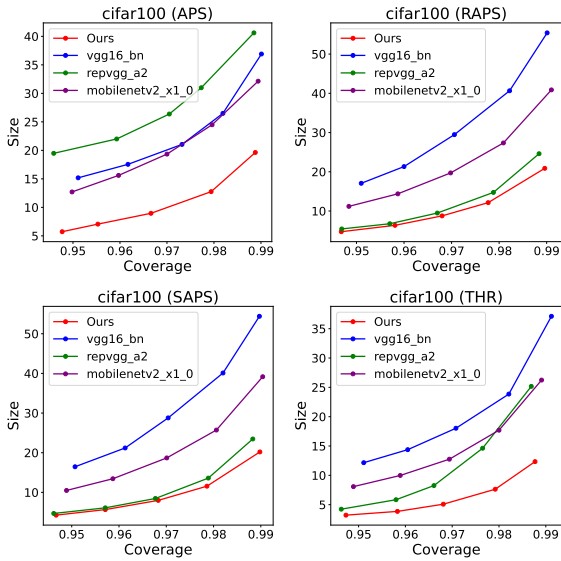

*Figure 5.* Across various score functions, our weighted combination of models outperformed any individual model and achieved optimal size on the CIFAR-100 dataset across $\alpha$ values (0.01–0.05).

subgraph theory techniques as in our method. Additionally, the model aggregation methods in (Carlsson et al., 2014; Linusson et al., 2017) can combine predictions from multiple models for conformal prediction, but they do not emphasize optimizing efficiency.

A concurrent preprint (Liang et al., 2024) examines parametrized score functions $s_\lambda(x, y)$ for $\lambda \in \Lambda$ within regression settings. While their general framework supports arbitrary parameter spaces, it faces challenges in controlling Rademacher complexity, except in certain specialized cases. In contrast, our approach linearly combines predefined score functions, introducing a structural constraint that allows theoretical guarantees to depend solely on the number of constituent scores, rather than the complexity of the function class.

## 6. Conclusion and Discussion

Our weighted score aggregation method enables efficient and valid prediction set construction for multi-class classification through optimized combinations of score functions and strategic data splitting. Theoretically, we establish finite-sample coverage guarantees and oracle inequalities quantifying the efficiency gap between our method and optimal weights. Empirically, experiments demonstrate consistent maintenance of coverage requirements with minimal prediction set sizes compared to single-score baselines. This work bridges model averaging and conformal prediction, providing a flexible framework for uncertainty quantification that adapts to dataset characteristics through optimal

score combinations.

These results suggest several directions for future work:

1. **Regression Extension**: Algorithmic adaptation to regression requires new theoretical tools due to continuous output spaces, where finite-class union bounds become inapplicable. Potential approaches include metric entropy analysis or covering number techniques.

2. **Optimization Enhancement**: Developing gradient-based alternatives to grid search, such as differentiable conformal objectives or online weight adaptation during model training, would improve scalability with many score functions.

## Acknowledgment

This work was partially supported by Hong Kong RGC and City University of Hong Kong grants (Project No. 9610639 and 6000864), DFG grant No. 389792660, and VolkswagenStiftung Grant AZ 98514. Zhixin Zhou's research was supported by the Genesis Award for Scientific Breakthrough from Alpha Benito LLC.

## Impact Statement

This work contributes to the broader goal of improving machine learning models' reliability and uncertainty quantification, which has the potential for positive societal impact across various domains.

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

## A. Grid Search Implementation

To solve (4), we discretize the probability simplex:

$$\Delta^{d-1} = \left\{ \boldsymbol{w} \in \mathbb{R}_+^d : \sum_{j=1}^d w_j = 1 \right\}$$

using a grid resolution $\varepsilon = 0.01$. Candidate weights are generated as:

$$\mathcal{W} = \left\{ \boldsymbol{w} = (k_1\varepsilon, \ldots, k_d\varepsilon) \;\middle|\; k_j \in \mathbb{N}, \; \sum_{j=1}^d k_j = \lceil 1/\varepsilon \rceil \right\},$$

yielding $\binom{\lceil 1/\varepsilon \rceil + d - 1}{d - 1}$ distinct weight vectors. For $d = 3$ scores, this produces 5,151 candidates, ensuring comprehensive coverage of the parameter space while remaining computationally tractable through parallel evaluation.

## B. Proof of Lemma 1

### B.1. VC Dimension of the Subgraph Classes

**Proposition 1.** *Both of the following classes of functions*

$$\{(x, y) \mapsto \mathbf{1}\{\langle \boldsymbol{w}, s(x, y)\rangle \geq t\} : \boldsymbol{w} \in \mathbb{R}^d, t \in \mathbb{R}\} \tag{9}$$

*and*

$$\{x \mapsto \mathbf{1}\{\langle \boldsymbol{w}, s(x, y)\rangle \geq t\} : \boldsymbol{w} \in \mathbb{R}^d, t \in \mathbb{R}\}, \tag{10}$$

*where $y \in [K]$ is fixed, have VC-dimension at most $d + 1$.*

*Proof.* Both of the classes are the subgraph classes of vector space of functions with dimension $d + 1$. By Proposition 4.20 of (Wainwright, 2019), these subgraph classes have dimension at most $d + 1$. This is also a direct result of Example 4.21 in the book. $\square$

### B.2. Proof of the Lemma

This proof basically follows from Vapnik–Chervonenkis theory. We will use the theorems and lemmas in (Wainwright, 2019) as reference. The proof of both parts of the lemma are almost identical. We will focus on part (a). Let $\epsilon_i, i \in \mathcal{I}$ be i.i.d. symmetric random variables take value $-1$ or $1$, i.e., $\mathbb{P}(\epsilon_i = -1) = \mathbb{P}(\epsilon_i = 1) = 0.5$. Then we define the *Rademacher complexity* for the function class in (9),

$$\mathcal{R}(\mathcal{I}) := \mathbb{E}\left[ \sup_{w,t} \left| \frac{1}{|\mathcal{I}|} \sum_{i \in \mathcal{I}} \epsilon_i \mathbf{1}\{\langle w, s(X_i, Y_i)\rangle \geq t\} \right| \right].$$

By Theorem 4.10 in the book,

$$\mathbb{P}\left(\Omega\left(\mathcal{I}, 2\mathcal{R}(\mathcal{I}) + \delta\right)\right) \geq 1 - \exp\left(-\frac{|\mathcal{I}|\delta^2}{2}\right). \tag{11}$$

By Proposition 1, the VC dimension of the function class in (9) is $d + 1$. Then altogether with Lemma 4.14 and Proposition 4.18 in the book,

$$\mathcal{R}(\mathcal{I}) \leq 4\sqrt{\frac{(d + 1)\log(|\mathcal{I}| + 1)}{|\mathcal{I}|}} \tag{12}$$

We combine (11) and (12) to obtain the result of part (a). The proof of part (b) of the lemma follows analogously, using the VC dimension result for the function class in (10). Let us define

$$\Gamma_y(\mathcal{I}, \eta) = \sup_{w \in \mathbb{R}^d, t \in \mathbb{R}} \left| \frac{1}{|\mathcal{I}|} \sum_{i \in \mathcal{I}} \mathbf{1}\{\langle w, s(X_i, y)\rangle \geq t\} - \mathbb{E}[\mathbf{1}\{\langle w, s(X, y)\rangle \geq t\}] \right| \leq \eta.$$

For $\mathcal{I} \subseteq \mathcal{I}_{\text{train}} \cup \mathcal{I}_{\text{test}}$,

$$\mathbb{P}\left(\Gamma_y\left(\mathcal{I}, 8\sqrt{\frac{(d+1)\log(|\mathcal{I}|+1)}{|\mathcal{I}|}} + \delta\right)\right) \geq 1 - \exp\left(-\frac{|\mathcal{I}|\delta^2}{2}\right).$$

It is clear that

$$\bigcup_{y \in [K]} \Gamma_y(\mathcal{I}, \eta) \subseteq \Gamma(\mathcal{I}, K\eta).$$

Taking the union bound on $y \in [K]$ implies the result of part (b).

## C. Preliminary Lemmas for the Theorems

### C.1. Bounds for Coverage Probability

The following result requires the data for optimizing $\widehat{w}$ are independent with the calibration set and the test set. The algorithm satisfying these conditions have the most reliable coverage probability.

**Lemma 2.** *Let $\widehat{C}_{1-\alpha}(x)$ be the output of Algorithm 2. Suppose*

(i) $\{(X_i, Y_i)\}_{i \in \mathcal{I}_3 \cup \mathcal{I}_{\text{test}}}$ *are exchangeable.*

(ii) $\{(X_i, Y_i)\}_{i \in \mathcal{I}_1 \cup \mathcal{I}_2}$ *and* $\{(X_i, Y_i)\}_{i \in \mathcal{I}_3 \cup \mathcal{I}_{\text{test}}}$ *are independent.*

*Then for $(X, Y)$ in the test set, the coverage probability*

$$\mathbb{P}(Y \in \widehat{C}_{1-\alpha}(X)) \geq 1 - \alpha.$$

*Proof.* $\widehat{w}$ only depends on $\{(X_i, Y_i)\}_{i \in \mathcal{I}_1 \cup \mathcal{I}_2}$, so it is independent of $\{(X_i, Y_i)\}_{i \in \mathcal{I}_3 \cup \mathcal{I}_{\text{test}}}$. The weighted score function $\{\langle \widehat{w}, s(X_i, Y_i)\rangle\}_{i \in \mathcal{I}_3 \cup \mathcal{I}_{\text{test}}}$ are also exchangeable. For any $(X, Y)$ in the test set, the rank of $s(X, Y)$ is smaller than $q_{1-\alpha}(\widehat{w}, \mathcal{I}_3)$ with probability $\frac{\lceil (1+|\mathcal{I}_3|)(1-\alpha)\rceil}{1+|\mathcal{I}_3|} \geq 1 - \alpha$. □

**Lemma 3.** *Let $\widehat{C}_{1-\alpha}(x)$ be the output of Algorithm 2. Suppose*

(i) $\Omega(\mathcal{I}_3, \eta_3)$ *is satisfied.*

(ii) $\{(X_i, Y_i)\}_{i \in \mathcal{I}_1 \cup \mathcal{I}_2}$ *and* $\{(X_i, Y_i)\}_{i \in \mathcal{I}_{\text{test}}}$ *are independent.*

*Then for $(X, Y)$ in the test set, the coverage probability*

$$\mathbb{P}(Y \in \widehat{C}_{1-\alpha}(X)) \geq \frac{1}{|\mathcal{I}_3|}\lceil (1 + |\mathcal{I}_3|)(1-\alpha)\rceil - \eta_3.$$

*Proof.* Let us write the threshold in (5) $q_{1-\alpha}^{(2)} := q_{1-\alpha}(\widehat{w}, \mathcal{I}_3)$ to emphasize that this quantile depends on $\widehat{w}$ and the samples in $\mathcal{I}_3$. By the procedure of the algorithm, we have

$$\mathbb{P}\left(Y \in \widehat{C}_{1-\alpha}(X)\right) = \mathbb{P}\left(\langle \widehat{w}, s(X, Y)\rangle \geq q_{1-\alpha}(\widehat{w}, \mathcal{I}_3)\right).$$

Assuming the event $\Omega(\mathcal{I}_3, \eta_3)$,

$$
\begin{aligned}
q_{1-\alpha}(\widehat{\boldsymbol{w}}, \mathcal{I}_3) &= \sup \left\{ t \in \mathbb{R} : \frac{1}{|\mathcal{I}_3|} \sum_{i \in \mathcal{I}_3} \mathbf{1}\{\langle \widehat{\boldsymbol{w}}, s(x_i, y_i) \rangle \geq t\} \geq \frac{1}{|\mathcal{I}_3|} \lceil (1 + |\mathcal{I}_3|)(1 - \alpha) \rceil \right\} \\
&\geq \sup \left\{ t \in \mathbb{R} : \mathbb{P}(\langle \widehat{\boldsymbol{w}}, s(X, Y) \rangle \geq t) \geq \frac{1}{|\mathcal{I}_3|} \lceil (1 + |\mathcal{I}_3|)(1 - \alpha) \rceil - \eta_3 \right\} \\
&= Q_{1-\alpha'}(\widehat{\boldsymbol{w}}),
\end{aligned}
\tag{13}
$$

where $1 - \alpha' = \frac{1}{|\mathcal{I}_3|} \lceil (1 + |\mathcal{I}_3|)(1 - \alpha) \rceil - \eta_3$. By the definition of the quantile,

$$
\mathbb{P}(\langle \widehat{\boldsymbol{w}}, s(X, Y) \rangle \geq q_{1-\alpha}(\widehat{\boldsymbol{w}}, \mathcal{I}_3)) \geq \mathbb{P}(\langle \widehat{\boldsymbol{w}}, s(X, Y) \rangle \geq Q_{1-\alpha'}(\widehat{\boldsymbol{w}})) = 1 - \alpha'.
\tag{14}
$$

The proof is complete. $\qquad\square$

**Lemma 4.** *Let $\widehat{C}_{1-\alpha}(X; \widehat{\boldsymbol{w}})$ be the output of Algorithm 2. Suppose $\Omega(\mathcal{I}_3, \eta_3)$ and $\Omega(\mathcal{I}_{test}, \eta_{test})$ hold, then the coverage proportion satisfies*

$$
\frac{1}{|\mathcal{I}_{test}|} \sum_{i \in \mathcal{I}_{test}} \mathbf{1}\{y_i \in \widehat{C}(x_i)\} \geq \frac{1}{|\mathcal{I}_3|} \lceil (1 + |\mathcal{I}_3|)(1 - \alpha) \rceil - \eta_3 - \eta_{test}.
$$

*Proof.* Under the event $\Omega(\mathcal{I}_{test}, \eta_{test})$, for all $w \in \mathcal{W}$,

$$
\begin{aligned}
&\left| \frac{1}{|\mathcal{I}_{test}|} \sum_{i \in \mathcal{I}_{test}} \mathbf{1}\{\langle \widehat{\boldsymbol{w}}, s(X_i, Y_i) \rangle \geq q_{1-\alpha}(\widehat{\boldsymbol{w}}, \mathcal{I}_3)\} - \mathbb{E}_{(X,Y)}[\mathbf{1}\{\langle \widehat{\boldsymbol{w}}, s(X, Y) \rangle \geq q_{1-\alpha}(\widehat{\boldsymbol{w}}, \mathcal{I}_3)\}] \right| \\
&\leq \sup_{\boldsymbol{w} \in \mathbb{R}^d, t \in \mathbb{R}} \left| \frac{1}{|\mathcal{I}_{test}|} \sum_{i \in \mathcal{I}_{test}} \mathbf{1}\{\langle \boldsymbol{w}, s(X_i, Y_i) \rangle \geq t\} - \mathbb{E}_{(X,Y)}[\mathbf{1}\{\langle \boldsymbol{w}, s(X, Y) \rangle \geq t\}] \right| \leq \eta_{test},
\end{aligned}
$$

where $(X, Y)$ is an i.i.d. copy of $(X_i, Y_i)$ in the test set. In particular, we let $\boldsymbol{w} = \widehat{\boldsymbol{w}}$ and $t = q_{1-\alpha}(\widehat{\boldsymbol{w}}, \mathcal{I}_3)$, we have

$$
E[\mathbf{1}\{\langle \widehat{\boldsymbol{w}}, s(X, Y) \rangle \geq Q_{1-\alpha'}(\widehat{\boldsymbol{w}}, \mathcal{I}_3)\} \mid \widehat{\boldsymbol{w}}] = \mathbb{P}(\langle \widehat{\boldsymbol{w}}, s(X, Y) \rangle \geq q_{1-\alpha}(\widehat{\boldsymbol{w}}, \mathcal{I}_3) \mid \widehat{\boldsymbol{w}}) = 1 - \alpha.
$$

The remaining proof has similar argument as (13) and (14), and is omitted here. $\qquad\square$

### C.2. Bounds for Prediction Set Size

**Lemma 5.** *Suppose the samples in $\mathcal{I}_1$ and $\mathcal{I}_2$ satisfy $\Omega(\mathcal{I}_1, \eta_1)$ and $\Gamma(\mathcal{I}_2, \xi_2)$ respectively, and suppose $Q_{1-\alpha'}(\widehat{\boldsymbol{w}}, \mathcal{I}_1) \leq q_{1-\alpha}(\widehat{\boldsymbol{w}}, \mathcal{I}_3)$, then for $X$ in the test set, the expected prediction set size satisfies*

$$
\frac{1}{|\mathcal{I}_2|} \sum_{i \in \mathcal{I}_2} \sum_{y \in [K]} \mathbf{1}\{\langle \widehat{\boldsymbol{w}}, s(X_i, y) \rangle \geq q_{1-\alpha}(\widehat{\boldsymbol{w}}, \mathcal{I}_3)\} \leq \mathbb{E}\left[ \sum_{y \in [K]} \mathbf{1}\{\langle w^*, s(X, y) \rangle \geq Q_{1-\alpha_1}(w^*)\} \right] + \xi_2,
$$

*where $1 - \alpha_1 = \frac{1}{|\mathcal{I}_1|} \lceil (1 + |\mathcal{I}_1|)(1 - \alpha') \rceil + \eta_1$.*

*Proof.* Under the assumption $Q_{1-\alpha'}(\widehat{\boldsymbol{w}}, \mathcal{I}_1) \leq q_{1-\alpha}(\widehat{\boldsymbol{w}}, \mathcal{I}_3)$, for $w \in \mathcal{W}$, we have

$$
\frac{1}{|\mathcal{I}_2|} \sum_{i \in \mathcal{I}_2} \sum_{y \in [K]} \mathbf{1}\{\langle w, s(X_i, y) \rangle \geq q_{1-\alpha}(w, \mathcal{I}_3)\} \leq \frac{1}{|\mathcal{I}_2|} \sum_{i \in \mathcal{I}_2} \sum_{y \in [K]} \mathbf{1}\{\langle w, s(X, y) \rangle \geq Q_{1-\alpha'}(w, \mathcal{I}_1)\}.
$$

$\widehat{\boldsymbol{w}}$ is obtained from

$$
\widehat{\boldsymbol{w}} \in \arg\min_{w \in \mathcal{W}} \frac{1}{|\mathcal{I}_2|} \sum_{i \in \mathcal{I}_2} \sum_{y \in [K]} \mathbf{1}\{\langle w, s(X, y) \rangle \geq Q_{1-\alpha'}(w, \mathcal{I}_1)\}
$$

Therefore,

$$\frac{1}{|\mathcal{I}_2|} \sum_{i \in \mathcal{I}_2} \sum_{y \in [K]} \mathbf{1}\{\langle \widehat{\boldsymbol{w}}, s(X_i, y) \rangle \geq Q_{1-\alpha'}(\widehat{\boldsymbol{w}}, \mathcal{I}_1)\} \leq \frac{1}{|\mathcal{I}_2|} \sum_{i \in \mathcal{I}_2} \sum_{y \in [K]} \mathbf{1}\{\langle w^*, s(X, y) \rangle \geq Q_{1-\alpha'}(w^*, \mathcal{I}_1)\}.$$

Given the event $\Omega(\mathcal{I}_1, \eta_1)$, for $w \in \mathcal{W}$ and $t \in \mathbb{R}$,

$$\mathbb{P}(\langle w, s(X, Y) \rangle \geq t) = \mathbb{E}[\mathbf{1}\{\langle w, s(X, Y) \rangle \geq t\}] \leq \frac{1}{|\mathcal{I}_1|} \sum_{i \in \mathcal{I}_1} \mathbf{1}\{\langle w, s(X, Y) \geq t\} + \eta_1.$$

The lower bound of the LHS is also a lower bound of the RHS, so we have

$$\begin{aligned}
Q_{1-\alpha'}(w^*, \mathcal{I}_1) &= \sup \left\{ t \in \mathbb{R} : \frac{1}{|\mathcal{I}_1|} \sum_{i \in \mathcal{I}_1} \mathbf{1}\{\langle w, s(x_i, y_i) \rangle \geq t\} \geq \frac{1}{|\mathcal{I}_1|} \lceil (1 + |\mathcal{I}_1|)(1 - \alpha') \rceil \right\} \\
&\geq \sup \left\{ t \in \mathbb{R} : \mathbb{P}(\langle w^*, s(X, Y) \rangle \geq t) \geq \frac{1}{|\mathcal{I}_1|} \lceil (1 + |\mathcal{I}_1|)(1 - \alpha') \rceil + \eta_1 \right\} \\
&= Q_{1-\alpha_1}(w^*),
\end{aligned} \tag{15}$$

where $1 - \alpha_1 = \frac{1}{|\mathcal{I}_1|} \lceil (1 + |\mathcal{I}_1|)(1 - \alpha') \rceil + \eta_1$ and $Q_{1-\alpha_1}(\boldsymbol{w}^*)$ is the $(1 - \alpha_1)$-quantile of the population distribution $\langle w, s(X, Y) \rangle$. Now we can conclude that

$$\frac{1}{|\mathcal{I}_2|} \sum_{i \in \mathcal{I}_2} \sum_{y \in [K]} \mathbf{1}\{\langle w^*, s(X, y) \rangle \geq Q_{1-\alpha'}(w^*, \mathcal{I}_1)\} \leq \frac{1}{|\mathcal{I}_2|} \sum_{i \in \mathcal{I}_2} \sum_{y \in [K]} \mathbf{1}\{\langle w^*, s(X, y) \rangle \geq Q_{1-\alpha_1}(w^*)\}.$$

On the event $\Gamma(\mathcal{I}_2, \xi_2)$,

$$\frac{1}{|\mathcal{I}_2|} \sum_{i \in \mathcal{I}_2} \sum_{y \in [K]} \mathbf{1}\{\langle w^*, s(X, y) \rangle \geq Q_{1-\alpha'}(w^*)\} \leq \mathbb{E}\left[ \sum_{y \in [K]} \mathbf{1}\{\langle w^*, s(X, y) \rangle \geq Q_{1-\alpha_1}(w^*)\} \right] + \xi_2.$$

The proof is complete. $\qquad\square$

## D. Proof of the Theorems

### D.1. Proof of Theorem 1

Under the assumption of the theorem, the setting of VFCP satisfies the condition of Lemma 2. This proves the coverage probability in the theorem. For the expected prediction set size, under the event $\Gamma(\mathcal{I}_2, \xi_2)$, by Lemma 5, for $X$ in the test set,

$$\mathbb{E}\left[ \sum_{y \in [K]} \mathbf{1}\{\langle \widehat{\boldsymbol{w}}, s(X_i, y) \rangle \geq q_{1-\alpha}(\boldsymbol{w}, \mathcal{I}_3)\} \right] \leq \frac{1}{|\mathcal{I}_2|} \sum_{i \in \mathcal{I}_2} \sum_{y \in [K]} \mathbf{1}\{\langle \widehat{\boldsymbol{w}}, s(X_i, y) \rangle \geq q_{1-\alpha}(\boldsymbol{w}, \mathcal{I}_3)\} + \xi_2.$$

This implies the coverage probability of the theorem.

### D.2. Proof of Theorem 2

Under the assumption of the theorem, the setting of VFCP satisfies the condition of Lemma 3. This proves the coverage probability in the theorem. For the expected prediction set size, since $\mathcal{I}_1 = \mathcal{I}_3$ and $\Gamma(\mathcal{I}_2, \xi_2)$ is satisfied, by Lemma 5, for $X$ in the test set,

$$\mathbb{E}\left[ \sum_{y \in [K]} \mathbf{1}\{\langle \widehat{\boldsymbol{w}}, s(X_i, y) \rangle \geq q_{1-\alpha}(\boldsymbol{w}, \mathcal{I}_3)\} \right] \leq \frac{1}{|\mathcal{I}_2|} \sum_{i \in \mathcal{I}_2} \sum_{y \in [K]} \mathbf{1}\{\langle \widehat{\boldsymbol{w}}, s(X_i, y) \rangle \geq q_{1-\alpha}(\boldsymbol{w}, \mathcal{I}_3)\} + \xi_2.$$

This implies the coverage probability of the theorem.

### D.3. Proof of Theorem 3

Since $\mathcal{I}_1 = \mathcal{I}_3 = \mathcal{I}_{\text{train}}$ and $\mathcal{I}_2 = \mathcal{I}_{\text{test}}$, one can verify that the theorem is the direct result of Lemma 4 and Lemma 5.

### D.4. Proof of Theorem 4

Similar as the setting of DLCP, the coverage probability is the direct result of Lemma 4. For the prediction set size, suppose $\Gamma(\mathcal{I}_2, \xi_2)$ holds, then for $X$ in the test set,

$$\mathbb{E}[|\widehat{C}_{1-\alpha}^{\text{DLCP+}}(X)|] \leq \frac{1}{|\mathcal{I}_2|} \sum_{i \in \mathcal{I}_2} \sum_{y \in [K]} \mathbf{1}\{\langle \widehat{\boldsymbol{w}}, s(X_i, y)\rangle \geq q_{1-\alpha}(\widehat{\boldsymbol{w}}, \mathcal{I}_3)\} + \xi_2$$

$$\leq \mathbb{E}\left[\sum_{y \in [K]} \mathbf{1}\{\langle w^*, s(X, y)\rangle \geq Q_{1-\alpha'}(w^*)\}\right] + 2\xi_2.$$

## E. Additional Discussion on Score Functions

The three basic score functions used in our experiment were:

**Least Ambiguous Set Values Classifier (THR) (Sadinle et al., 2019).** The score function of THR is defined as:

$$s_{\text{THR}}(x, y) = \widehat{p}_y(x).$$

This score function is straightforward: it assigns a higher score to labels with a higher estimated probability. The prediction set includes labels with the highest estimated probabilities. This is the score function we have to include because if $\widehat{p}$ is the true posterior probability, then the score function itself can achieve the smallest expected prediction set size.

**Adaptive Prediction Set (APS) (Romano et al., 2020).** The score function of APS is defined as:

$$s_{\text{APS}}(x, y) = \sum_{y' \in [K]} \widehat{p}_{y'}(x)\mathbf{1}\{\widehat{p}_{y'}(x) \leq \widehat{p}_y(x)\},$$

where $\mathbf{1}\{\cdot\}$ is the indicator function. This score function can be interpreted as the complement of the $p$-value for label $k$. It measures the sum of the estimated probabilities of all labels that have the same or a smaller estimated probability than label $k$.

**Rank-based Score Function (RANK) (Luo & Zhou, 2024).** The score function of RANK is defined as:

$$s_{\text{RANK}}(x, y) = \frac{|\{k' \in [K] : \widehat{p}_{k'}(x) < \widehat{p}_k(x)\}|}{K - 1},$$

This score function assigns a score based on the rank of the estimated probability $\widehat{p}_y(x)$ among all the estimated probabilities for input $x$. The rank is divided by $K - 1$ so that the range of the score is from 0 to 1. The prediction set gives higher priority to labels with larger ranks.

The three score functions, THR, APS, and RANK, employ different strategies for assigning scores based on the estimated probabilities. THR directly utilizes the estimated probabilities as scores. APS, on the other hand, considers the cumulative probability of labels that have the same or lower estimated probabilities compared to the label of interest. RANK, in contrast, focuses on the relative ranking of the estimated probabilities among all possible labels. It is important to note that all these score functions preserve the order of the labels, which means that, for fixed $x$, the order of labels based on their estimated probabilities, $\widehat{p}_y(x)$, remains the same when ranked according to their scores, $s(x, y)$.

# F. Additional Experiments with Different Datasets

We conducted additional experiments on MNIST, Fashion-MNIST, and ImageNet-Val, including comparisons with the Synergy Conformal Prediction (SCP) method (Gauraha & Spjuth, 2021), suggested by Reviewer bRdZ. Each experiment used 2000 samples, 100 runs with different calibration/test splits (as in Table 1, Section 4.1), the APS score function, and the EFCP split method. Results show our method consistently achieves smaller prediction sets while maintaining coverage at $\alpha = 0.01$ and $\alpha = 0.05$.

## F.1. MNIST

| Method | Coverage ($\alpha = 0.01$) | Size ($\alpha = 0.01$) | Coverage ($\alpha = 0.05$) | Size ($\alpha = 0.05$) |
|---|---|---|---|---|
| Ours | 0.988 (0.005) | **1.577 (0.061)** | 0.951 (0.011) | **1.001 (0.011)** |
| SVM | 0.990 (0.005) | 2.323 (0.135) | 0.950 (0.011) | 1.033 (0.014) |
| Random Forest | 0.990 (0.005) | 2.205 (0.108) | 0.951 (0.013) | 1.181 (0.023) |
| Logistic Regression | 0.990 (0.005) | 3.695 (0.123) | 0.950 (0.012) | 1.557 (0.065) |
| SCP | 0.990 (0.005) | 1.771 (0.083) | 0.951 (0.012) | 1.018 (0.011) |

*Table 2.* Results on MNIST dataset.

## F.2. Fashion-MNIST

| Method | Coverage ($\alpha = 0.01$) | Size ($\alpha = 0.01$) | Coverage ($\alpha = 0.05$) | Size ($\alpha = 0.05$) |
|---|---|---|---|---|
| Ours | 0.988 (0.006) | **2.296 (0.108)** | 0.948 (0.012) | **1.265 (0.031)** |
| SVM | 0.991 (0.005) | 2.941 (0.156) | 0.952 (0.012) | 1.449 (0.044) |
| Random Forest | 0.990 (0.006) | 3.264 (0.187) | 0.949 (0.012) | 1.612 (0.035) |
| Logistic Regression | 0.989 (0.006) | 3.325 (0.120) | 0.949 (0.012) | 1.841 (0.050) |
| SCP | 0.991 (0.005) | 2.446 (0.100) | 0.951 (0.013) | 1.315 (0.030) |

*Table 3.* Results on Fashion-MNIST dataset.

## F.3. ImageNet-Val

| Method | Coverage ($\alpha = 0.01$) | Size ($\alpha = 0.01$) | Coverage ($\alpha = 0.05$) | Size ($\alpha = 0.05$) |
|---|---|---|---|---|
| Ours | 0.989 (0.006) | **48.264 (4.019)** | 0.949 (0.013) | **6.670 (0.618)** |
| ResNet101 | 0.990 (0.005) | 53.744 (3.940) | 0.950 (0.013) | 6.798 (0.631) |
| VGG16 | 0.991 (0.005) | 100.683 (9.445) | 0.950 (0.011) | 15.149 (0.831) |
| ResNet18 | 0.990 (0.006) | 110.401 (11.694) | 0.949 (0.011) | 18.323 (1.422) |
| SCP | 0.991 (0.005) | 77.243 (5.667) | 0.950 (0.011) | 10.661 (0.630) |

*Table 4.* Results on ImageNet-Val dataset.

# G. Additional Experiments with Different Data Splitting Ratio

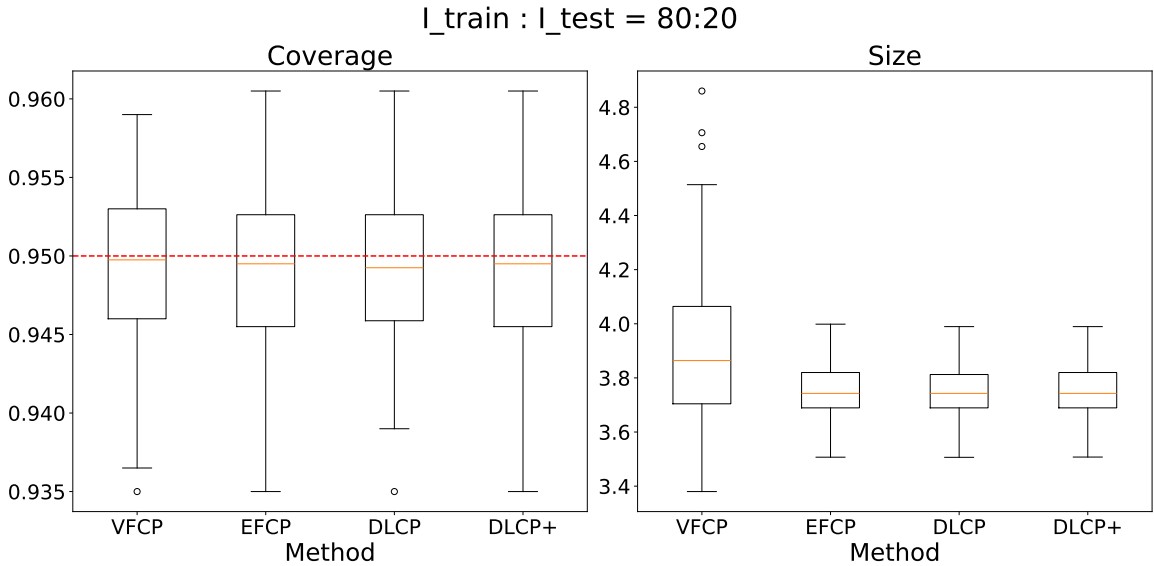

(a) Comparison of coverage and size for different data split methods at a significance level of $\alpha = 0.05$ when $\mathcal{I}_{\text{train}} : \mathcal{I}_{\text{test}} = 80:20$. EFCP, DLCP, and DLCP+ exhibit similar size results, but DLCP has the smallest coverage and the largest gap from the desired coverage level of $1 - \alpha = 0.95$. VFCP attains the desired coverage at the cost of having the largest prediction set size.

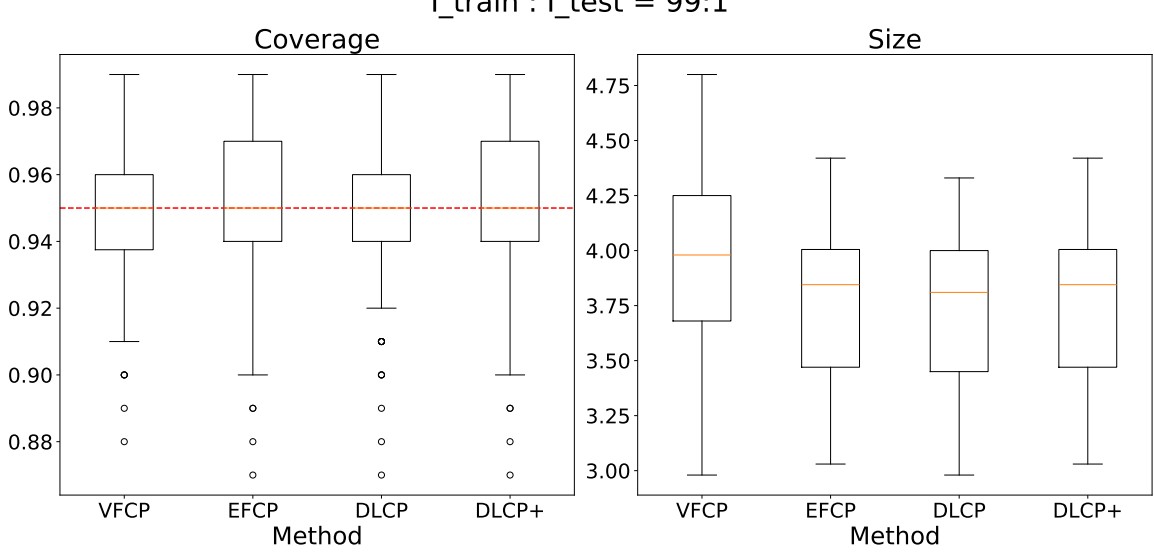

(b) Comparison of coverage and size for different data split methods at a significance level of $\alpha = 0.05$ when $\mathcal{I}_{\text{train}} : \mathcal{I}_{\text{test}} = 99:1$. DLCP achieves the smallest size among the compared methods. VFCP attains the desired coverage at the cost of having the largest prediction set size.

*Figure 6.* Comparison of coverage and size for different data split methods at $\alpha$=0.05.

