# OpenReview forum: "Conformity Score Averaging for Classification"
_ICML.cc/2025/Conference — ICML 2025 poster_

### Official Review · Reviewer_bRdZ · 2025-03-05

**Overall Recommendation:** 3

**Summary:**

This paper proposes to improve conformal prediction by optimally averaging multiple conformity score functions. This papers explores various data splitting methods and optimal weights for aggregating the score functions.

**Claims And Evidence:**

The main claim of this paper is that by optimally weighting multiple score functions, the resulting threshold can achieve better performance.   This claim is validated both theoretically and numerically. However, in terms of theoretical results, this paper shows that optimizing weights has small impact on the validity. However, the validity of proposed threshold does not directly support the claim of improving performance. The numerical experiments are limited to two datasets.

**Essential References Not Discussed:**

Gauraha, Niharika, and Ola Spjuth. "Synergy conformal prediction." Conformal and Probabilistic Prediction and Applications. PMLR, 2021.

**Experimental Designs Or Analyses:**

The experiment metric makes sense, but it lacks a comprehensive comparison with other alternative ensemble methods.

**Methods And Evaluation Criteria:**

* The proposed method makes sense but lacks principled guidance.
  * Ensemble method is common in machine learning. Despite the claim, it is unclear where the novelty lies.
  * The optimal weights are determined by grid search, which is potentially very inefficient. Instead of investigating further on the optimal weights, this paper go into different splitting methods, which blur the main focus and the main take away on different splitting is also not clear.
* In terms of evaluation, the benchmark datasets are not sufficient and missing out on others like variants of MNIST.

**Other Comments Or Suggestions:**

N/A

**Other Strengths And Weaknesses:**

N/A

**Questions For Authors:**

N/A

**Relation To Broader Scientific Literature:**

N/A

**Theoretical Claims:**

Limited review. Appears to be correct in terms of statements. But it lacks interpretation in the main texts of the paper.

---

> ### Author Rebuttal · Authors · 2025-03-30
>
> Thank you for your detailed feedback and constructive suggestions.
>
> ---
>
> ### **1. About why our method can improve performance**
>
> In our setting, there exist $d$ score functions corresponding to weights $w = e_1, \dots, e_d$, where $e_i$ is the $i$-th standard basis in $\mathbb{R}^d$. The optimal weight $w^* \in \Delta^d$ yields the most efficient score function. Since $\{e_1, \dots, e_d\} \subset \Delta^d$, the score function $\langle w^*, s(x, y) \rangle$ is at least as efficient as any single score function. The improvement of our method is quantified by the difference in prediction set size between $\langle w^*, s(x, y) \rangle$ and the best single-score function.
>
> ---
>
> ### **2. About novelty in the ensemble method literature**
>
> We agree that ensemble methods are common in machine learning and statistics, and this motivates our proposal to apply model averaging to conformity scores.
>
> Unlike standard ensemble methods, our approach aggregates scores in the final step during conformal prediction. This allows ensemble methods to be applied even when only one model is trained, by aggregating different score functions. To the best of our knowledge, this is the first work that enables ensemble techniques at the conformity score level.
>
> Theoretically, our paper is the first to show that averaging conformity scores has a mild effect on validity while achieving near-optimal efficiency.
>
> ---
>
> ### **3. About inefficiency of grid search**
>
> We acknowledge that grid search may not be efficient and view efficient weight optimization as future work. We have introduce some alternative method in the response to Reviewer j2fr and Reviewer gF9v. The methods include greedy search, golden section search, gradient descent after smoothing and stochastic optimization.
>
> While we acknowledge the importance of efficient optimization, the focus of this work is on analyzing the properties of $\hat{w}$ and providing its statistical guarantees.
>
> ---
>
> ### **4. About why we consider different splitting methods**
>
> We respectfully disagree with the assertion that introducing different splitting methods blurs the main focus of the paper. As described in Section 2.5, we use only 34 lines to introduce these methods, which we believe is a reasonable length. It is necessary to present the splitting methods here, as the statistical guarantees in the next section depend on them.
>
> At least one splitting method is essential for finding $\hat{w}$. The main takeaway from introducing different splitting methods is to demonstrate that our conformity score averaging method is versatile and can be applied in various splitting scenarios.
>
> As pointed out by Reviewer gF9v, examining a variety of splits and clarifying how coverage versus efficiency tradeoffs shift is one of the strengths of our work. Therefore, we will retain the splitting methods, along with their theoretical analysis and experiments, in the paper.
>
> ---
>
> ### **5. About lack of datasets, such as variants of MNIST**
>
> We have conducted additional experiments to address this concern. Please see the updated experimental results in our response to Reviewer 82f9.
>
> ---
>
> ### **6. About interpretation of theoretical claims**
>
> Thank you for confirming the correctness of our theoretical results. Section 3 provides a detailed interpretation of our theorems, with the key takeaway being that $\eta$ and $\xi$ are small as long as the dataset sizes are large.
>
> As suggested by Reviewer 82f9, we will add additional intuition about the VC dimension and its connection to the DKW inequality to make the theoretical results more accessible.
>
> ---
>
> ### **7. About comparison with other alternative ensemble methods**
>
> The most relevant ensemble method is in [1], which outputs the most efficient prediction set among a finite set of score functions. Our experiments already compare against single scores and models, including the most efficient ones. Please see Table 1 and Figures 3–5, where our method consistently outperforms these baselines.
>
> ---
>
> ### **8. About the suggested reference, "Synergy conformal prediction"**
>
> Thank you for suggesting this reference. This paper also proposes an ensemble method for conformal prediction. However, our method differs in two key ways:
>
> 1. Our averaging occurs at the conformity score level, while their method aggregates models.
> 2. Our method optimizes the weights, whereas theirs does not.
>
> We will cite this paper in the revised manuscript and clarify the distinctions between the two methods. Please see the experiment result in our response to Reviewer 82f9.
>
> ---
>
> ### **References**
>
> [1] Yang, Y., and Kuchibhotla, A. K. (2024). Selection and aggregation of conformal prediction sets. *Journal of the American Statistical Association*, 1–13.

---

> > ### Comment · Reviewer_bRdZ · 2025-04-05
> >
> > I appreciate the authors’ detailed and helpful response. The added clarifications improved the theoretical exposition, and the new experimental results demonstrate the benefits of the proposed method. Overall, I find the paper more convincing and have increased my score accordingly. One last thing I wanted to mention is that a brief discussion or recommendation on selecting data splitting strategies could provide more practical value.

---

### Official Review · Reviewer_gF9v · 2025-03-14

**Overall Recommendation:** 4

**Summary:**

The paper proposes a method for improving prediction set efficiency in classification tasks through conformity score averaging. It introduces weighted averaging of multiple score functions and explores various data-splitting strategies to optimize the weight selection process. Theoretical guarantees for coverage and efficiency are established using VC theory, and experiments on CIFAR-10 and CIFAR-100 datasets demonstrate the method’s effectiveness over existing approaches.

**Claims And Evidence:**

The claims are mostly supported by the evidence presented in the paper. More precisely:

- Averaging multiple conformity score functions leads to more efficient (i.e., smaller) prediction sets than using any single score alone.
- The method retains finite-sample coverage guarantees for multi-class classification.
- Empirical results on CIFAR-10 and CIFAR-100 show that the weighted method consistently achieves the target coverage while reducing prediction set size.
- Theoretical proofs based on VC theory support the coverage and efficiency claims.
- While convincing within the scope of the experiments, extending evaluations to more diverse or real-world datasets could further strengthen the evidence.

**Essential References Not Discussed:**

The related works in the paper is fine. While not strictly missing from the discussion, the following paper might be interesting to consider:

Gasparin, Matteo, and Aaditya Ramdas. "Merging uncertainty sets via majority vote." arXiv preprint arXiv:2401.09379 (2024).

**Experimental Designs Or Analyses:**

The experimental design and analysis is fine.

- Experiments conducted on benchmark datasets such as CIFAR-10 and CIFAR-100.
- Performance is compared against single-score methods (e.g., THR, APS, RANK) and competitive baselines (e.g., RAPS, SAPS).
- Multiple experimental runs (e.g., 100 different splits) and varying significance levels ($\alpha$ values).
- Key metrics include coverage probability and average prediction set size.
- Although the experiments on CIFAR datasets are well-conducted, incorporating additional datasets from different domains could enhance the findings.

**Methods And Evaluation Criteria:**

Yes, the proposed methods and evaluation criteria make sense. Some remarks in this regard:

- Weighted averaging of several nonconformity scores using a grid search over a discretized probability simplex.
- Implementation of different data splitting strategies (VFCP, EFCP, DLCP, DLCP+) to  optimize weight selection.
- The use of grid search over the probability simplex to determine optimal weights is  sensible, though computationally intensive.
- Evaluating the methods based on coverage probability and average prediction set size aligns well with the goals of both validity and efficiency in uncertainty quantification (and is standard practice in the CP literature).

**Other Comments Or Suggestions:**

Some suggestions:

- Implementations might consider more direct/continuous optimization of the average set size, e.g., using subgradients w.r.t $\omega$.
- Extending to the label-conditional coverage or group-conditional coverage would be an interesting next step.
- The authors might systematically measure coverage vs. set size as a function of the calibration set size or the fraction used for weight selection.

**Other Strengths And Weaknesses:**

Strengths:

- The linear combination of base scores is straightforward and can unify multiple advanced scoring rules.
- They rely on standard VC arguments to guarantee coverage and near-optimal size.
- They examine a variety of splits (VFCP, EFCP, DLCP, DLCP+), clarifying how coverage vs. efficiency can shift.
- In CIFAR tasks, the method consistently matches or outperforms single-score baselines, achieving the nominal coverage with smaller sets.


Weaknesses:

- The extension to regression remains open because the union-bounding trick used to handle classification does not trivially extend to real intervals (this limitation is acknowledged, though).
- The method does a full grid search over $\Delta^d$, which can be expensive if the number of base scores or the resolution is large.
- The paper’s experiments focus on CIFAR-10 and CIFAR-100. Additional or larger datasets (e.g., ImageNet) or real-world tasks could better test the method’s scalability.

**Questions For Authors:**

Questions:

-  Is it possible combine the data leakage idea with cross-validation or bootstrap to reduce coverage biases while still leveraging more data to pick $\hat{\omega}$?
- Have you considered applying a subgraph approach to real intervals for regression tasks? Perhaps bounding the capacity by restricting the family of linear transformations.

**Relation To Broader Scientific Literature:**

The work builds on the established framework of conformal prediction and extends it by integrating ideas from model averaging. It compares and contrasts with earlier methods such as APS, RAPS, and SAPS, highlighting improvements in efficiency.

**Theoretical Claims:**

I tried to check all theoretical claims (and proofs thereof) in the paper. However, I did not check the corresponding Appendix line by line (mostly the "critical" parts to make sure that everything aligns).

Remarks:

- Finite-sample coverage guarantees are established via a theoretical analysis grounded in VC theory.
- The paper provides oracle inequalities that quantify the efficiency gap between the proposed weighted method and the ideal (optimal) prediction set.
- Consistency results demonstrate that the prediction set size approaches that of the optimal predictor as the sample size increases.

---

> ### Author Rebuttal · Authors · 2025-03-30
>
> Thank you very much for your positive review. Most of your questions are high-level and insightful. We will address them in order, starting with simpler questions and moving to more complex, open-ended ones.
>
> ---
>
> ### **1. About additional experiments**
>
> We have conducted additional experiments on MNIST, Fashion-MNIST, and ImageNet-Val, as detailed in our response to Reviewer 82f9. These results will be included in the revised version of the paper.
>
> ---
>
> ### **2. Coverage vs. set size as a function of the calibration set size**
>
> If we understand the question correctly, this refers to the relationship between calibration set size, coverage accuracy, and prediction set size. We have not systematically measured this relationship as a function, as this type of experiment is computationally expensive. Such an experiment would require multiple repetitions (e.g., 100 runs) to observe meaningful differences.
>
> In Section F, we provide related experiments comparing two calibration set sizes. The results show that a larger calibration set provides more accurate coverage. However, the prediction set size also depends on the splitting method used.
>
> ---
>
> ### **3. About conditional coverage**
>
> Our method can be applied to conditional coverage. The current work focuses on improving the efficiency of conformal prediction, with the prediction set size as the key optimization criterion. However, this criterion can be adapted to metrics related to conditional coverage, such as group-conditional coverage, label-conditional coverage, or ECE. By adjusting the criterion and allowing the algorithm to search for $\hat{w}$ that optimizes it, our method can generate conformal prediction sets tailored to meet conditional coverage requirements.
>
> ---
>
> ### **4. About cross-validation or bootstrap**
>
> We do not believe the data leakage idea can be directly combined with cross-validation or bootstrap. The data leakage method uses the prediction set sizes of test samples to determine the weight $\hat{w}$. Below, we outline our thoughts on how cross-validation and bootstrap might connect to this idea.
>
> A generalized version of the main algorithm could be considered. In Algorithm 2, $\mathcal{I}_1$ is used to find $\hat{w}$, and $\mathcal{I}_2$ is used to determine the quantile corresponding to $\hat{w}$. In VFCP, $\mathcal{I}_1$ and $\mathcal{I}_2$ are fixed partitions. A possible generalization would involve sampling $\mathcal{I}_1$ and $\mathcal{I}_2$ multiple times (e.g., through cross-validation or bootstrap). This process would result in multiple estimates of $\hat{w}$. Averaging these estimates could yield a final $\hat{w}$. The question is do we need another dataset to determine the threshold. This brings us back to the choice between VFCP and EFCP: whether to perform calibration on the same set or a separate set.
>
> We do not have a clear theoretical answer on how these methods should be properly applied. We hypothesize that cross-validation or bootstrap could provide an intermediate solution between VFCP and EFCP.
>
> ---
>
> ### **5. About efficient methods to find $\hat{w}$**
>
> We have explored several methods to improve the efficiency of finding $\hat{w}$. Some simpler methods are introduced in our response to Reviewer gF9v.
>
> As you suggested, gradient descent is a potential approach. The main technical challenge is that the prediction set size is not a continuous function of $w$. To address this, we can approximate the indicator functions in the objective function using sigmoid functions as conformal training does. The success of this approach heavily depends on tuning the temperature parameter of the sigmoid function.
>
> We have also considered stochastic optimization. For instance, starting from a random point $w$ on the grid $\Delta^d$, we evaluate the prediction set size for $w$ and its neighbors. We then move to a neighboring point with a probability based on the prediction set size. We record the smallest prediction set size encountered and use the corresponding $w$ as the final $\hat{w}$. Alternatively, discrete steps can be replaced with normal variable on $\Delta^d$. This energy-based method has shown promising results in some experiments, and we plan to include it in future work.
>
> ---
>
> ### **6. About extension to regression problems**
>
> Our theoretical results can be extended to some families of score functions. Specifically, Lemma 1(b) relies on the VC dimension of the subgraph class:
> $$\mathcal{A}:= \\{ \\{ y : \langle w, s(x, y) \rangle \geq t \\} : w \in \mathbb{R}^d, t \in \mathbb{R} \\}.$$
> If the scores $s(x, y)$ are always concave in $y$, then the weighted score $\langle w, s(x, y) \rangle$ is also concave in $y$. In this case, all elements of $\mathcal{A}$ are intervals, and the VC dimension of $\mathcal{A}$ is 2. This result suggests that our method can handle any number of concave score functions, including those used in CQR. We believe this result will be useful to some readers and will include it in the revision.

---

### Official Review · Reviewer_j2fr · 2025-03-14

**Overall Recommendation:** 3

**Summary:**

In this paper, the authors presented an approach that enhances conformal prediction for multi-class classification by optimally averaging multiple conformity score functions, and a set of evaluation experiments showed that the weighted averaging approach consistently outperforms single-score methods by producing smaller prediction sets without sacrificing coverage.

**Claims And Evidence:**

A set of the experiments using CIFAR10 and CIFAR100 using different significance levels, and the performance comparisons
of various models and their weighted combinations, *empirically* supported the outperformance of using the optimally averaging approach.

**Essential References Not Discussed:**

Seems not applicable.

**Experimental Designs Or Analyses:**

Please refer to the questions in the Methods and Evaluation Criteria Section.

**Methods And Evaluation Criteria:**

Some questions are listed as follows:

Could the authors provide insights into the complexity comparison between the proposed approach and single-score methods? A clearer understanding would help the community assess the balance between computational overhead and performance improvement.

It is not clear in the main body how many runs are conducted for each experimental setting.

**Other Comments Or Suggestions:**

Additional comment: Impact Statement is missing in the submission.

**Other Strengths And Weaknesses:**

Additional strengths: The paper structure is well organised, the motivation is clearly described.

**Questions For Authors:**

No futher questions.

**Relation To Broader Scientific Literature:**

It looks fine.

**Theoretical Claims:**

Given that the presented research field extends beyond my area of expertise, thoroughly assessing its theoretical soundness in detail is challenging. My current evaluation may be conservative and I will make my best effort to actively engage in discussions with the authors, other reviewers, and ACs.

---

> ### Author Rebuttal · Authors · 2025-03-30
>
> Thank you for your positive feedback and for appreciating the organization and motivation of our paper. Below, we address your questions and comments.
>
> ---
>
> ### **1. About the complexity of the proposed approach**
>
> In our algorithm, the main source of complexity lies in finding the optimal weight $\hat{w}$. Specifically, we need to compute the prediction set size $O(1/\epsilon^d)$ times, where $\epsilon$ is the grid size, as detailed in Section A of the appendix.
>
> We emphasize that evaluating prediction sets is computationally efficient for each iteration. Although the complexity is exponential in $d$, the process of finding $\hat{w}$ is still significantly faster than training deep learning models for classification tasks.
>
> To address scalability when the number of score functions is small, we suggest the following greedy approach:
>
> 1. Start with $w_1 = 1$.
> 2. Add $w_2$ so that $w_1 s_1 + w_2 s_2$ minimizes the prediction set size.
> 3. Continue iteratively, adding $w_3, w_4, \dots$, ensuring each step minimizes the prediction set size.
>
> Additionally, we observed in our experiments that the prediction set size is often quasi-convex in $w_i$, allowing for optimization via the golden-section search method. However, we cannot guarantee convexity in general, meaning this greedy approach may not yield the theoretically optimal $\hat{w}$ or satisfy the statistical guarantees in Section 3.
>
> We discuss possible additional efficient methods in our response to Reviewer gF9v, who has suggested a useful approach. Kindly refer to that response for further details.
>
> The focus of this work is on analyzing the properties of $\hat{w}$, rather than developing efficient algorithms to compute it. We believe that designing efficient methods to find $\hat{w}$ is an exciting direction for future work.
>
> ---
>
> ### **2. About the number of runs in the experiments**
>
> If we understood your question correctly, the number of runs is 100. This is stated on line 368 of page 7. In each run, the data is randomly partitioned into training, calibration, and test sets.
>
> ---
>
> ### **3. About the impact statement**
>
> Thank you for reminding us about the impact statement. We apologize for its omission. We will add the following statement to the revised manuscript:
>
> *This work contributes to the broader goal of improving machine learning models' reliability and uncertainty quantification, which has the potential for positive societal impact across various domains.*

---

### Official Review · Reviewer_82f9 · 2025-03-17

**Overall Recommendation:** 3

**Summary:**

Existing conformal prediction methods typically rely on a single conformity score function, limiting both their efficiency and informativeness. In this paper, they propose a new approach that enhances conformal prediction by averaging multiple conformity score functions for the same classification task. They also provide a comprehensive theoretical analysis based on VC theory, demonstrating the effectiveness of our method.

**Claims And Evidence:**

The proposed research problem is clearly defined, and the discussion around using a single score function is valid. The approach of combining multiple score functions with data splitting to enhance conformal prediction is logically sound. Furthermore, the paper provides comprehensive theoretical support to demonstrate the advantages of the proposed method, and the experiments empirically validate its claims.

**Essential References Not Discussed:**

n.a.

**Experimental Designs Or Analyses:**

Yes. The overall experimental design makes sense, but expanding the experimental settings would make the work more solid.

**Methods And Evaluation Criteria:**

The proposed methods and evaluation criteria are logically well-founded for the stated problem.

**Other Comments Or Suggestions:**

see weakness

**Other Strengths And Weaknesses:**

Strengths:
1. The research problem is intriguing, as it explores how to apply multiple conformal score functions to a single classification task.
2. The proposed method is both intuitive and logically sound.
3. The theoretical analysis is thorough, providing a comprehensive basis for the method’s correctness.

Weaknesses
1. The rationale for using the VC dimension should be explained in a more intuitive manner, making it easier for readers to understand its relevance.
2. Emphasize how the proposed theoretical contributions surpass or improve upon existing methods, which will help highlight the significance of the work.
3. For the experimental evaluation, only two datasets were used, which may limit the persuasiveness of the results. Including additional datasets or varying the experimental settings would further strengthen the evidence for the proposed method.

**Questions For Authors:**

n.a.

**Relation To Broader Scientific Literature:**

Conformal prediction is a flexible technique that can be applied in a variety of domains. The methods proposed in this work further advance the development of conformal prediction.

**Theoretical Claims:**

Yes. The theoretical claims are comprehensive and are supported by a detailed proof.

---

> ### Author Rebuttal · Authors · 2025-03-30
>
> Thank you for your positive feedback and helpful suggestions. We also appreciate your careful review of the supplemental materials. Below, we will address your comments.
>
> ---
>
> ### **1. About the intuition of VC dimension in our proof**
>
> We agree that providing this intuition would benefit readers. The primary idea relies on the Dvoretzky–Kiefer–Wolfowitz (DKW) inequality. If the supremum in equation (7) or (8) is taken over a single variable, e.g., $q \in \mathbb{R}$ or $w_1 \in \mathbb{R}$, the DKW inequality can be directly applied.
>
> Using the VC dimension generalizes the DKW inequality, allowing the supremum to extend over multiple variables when the function is linear in those variables. This generalization underpins our proof's validity and efficiency. We will add this explanation to the revision.
>
> ---
>
> ### **2. About the theoretical contribution compared to existing papers**
>
> In Section 5, we compare our method to two relevant approaches and can elaborate as follows:
>
> - The method in [1] optimizes over $w \in \\{e_1, \dots, e_d\\}$, where $e_i$ is the $i$-th standard basis in $\mathbb{R}^d$. In contrast, we optimize over $w \in \Delta^d$, considering a much larger set of candidate score functions. Both methods show mild validity loss, but our approach is more efficient.
>
> - The concurrent preprint [2] defines score functions as $s_w(x, y)$ compared to our $\langle w, s(x, y) \rangle$. While both use Rademacher complexity to analyze validity and efficiency, [2] cannot provide implicit bounds for validity or prediction set size via subgraph theory, as we do. This highlights our unique theoretical contribution.
>
> ---
>
> ### **3. About experiments**
>
> We conducted additional experiments on MNIST, Fashion-MNIST, and ImageNet-Val, including comparisons with the Synergy Conformal Prediction (SCP) method [3], suggested by Reviewer bRdZ. Each experiment used 2000 samples, 100 runs with different calibration/test splits (as in Table 1, Section 4.1), the APS score function, and the EFCP split method. Results show our method consistently achieves smaller prediction sets while maintaining coverage at α= 0.01 and = 0.05.
>
> #### **MNIST**
> | Method                | Coverage (α=0.01) | Size (α=0.01) | Coverage (α=0.05) | Size (α=0.05) |
> |-----------------------|--------------------|---------------|--------------------|---------------|
> | **Ours**             | 0.988 (0.005)     | **1.577 (0.061)** | 0.951 (0.011)     | **1.001 (0.011)** |
> | SVM                  | 0.990 (0.005)     | 2.323 (0.135) | 0.950 (0.011)     | 1.033 (0.014) |
> | Random Forest        | 0.990 (0.005)     | 2.205 (0.108) | 0.951 (0.013)     | 1.181 (0.023) |
> | Logistic Regression  | 0.990 (0.005)     | 3.695 (0.123) | 0.950 (0.012)     | 1.557 (0.065) |
> | SCP                  | 0.990 (0.005)     | 1.771 (0.083) | 0.951 (0.012)     | 1.018 (0.011) |
>
> #### **Fashion-MNIST**
> | Method                | Coverage (α=0.01) | Size (α=0.01) | Coverage (α=0.05) | Size (α=0.05) |
> |-----------------------|--------------------|---------------|--------------------|---------------|
> | **Ours**             | 0.988 (0.006)     | **2.296 (0.108)** | 0.948 (0.012)     | **1.265 (0.031)** |
> | SVM                  | 0.991 (0.005)     | 2.941 (0.156) | 0.952 (0.012)     | 1.449 (0.044) |
> | Random Forest        | 0.990 (0.006)     | 3.264 (0.187) | 0.949 (0.012)     | 1.612 (0.035) |
> | Logistic Regression  | 0.989 (0.006)     | 3.325 (0.120) | 0.949 (0.012)     | 1.841 (0.050) |
> | SCP                  | 0.991 (0.005)     | 2.446 (0.100) | 0.951 (0.013)     | 1.315 (0.030) |
>
> #### **ImageNet-Val**
> | Method                | Coverage (α=0.01) | Size (α=0.01) | Coverage (α=0.05) | Size (α=0.05) |
> |-----------------------|--------------------|---------------|--------------------|---------------|
> | **Ours**             | 0.989 (0.006)     | **48.264 (4.019)** | 0.949 (0.013)     | **6.670 (0.618)** |
> | ResNet101            | 0.990 (0.005)     | 53.744 (3.940) | 0.950 (0.013)     | 6.798 (0.631) |
> | VGG16                | 0.991 (0.005)     | 100.683 (9.445) | 0.950 (0.011)     | 15.149 (0.831) |
> | ResNet18             | 0.990 (0.006)     | 110.401 (11.694) | 0.949 (0.011)     | 18.323 (1.422) |
> | SCP                  | 0.991 (0.005)     | 77.243 (5.667) | 0.950 (0.011)     | 10.661 (0.630) |
>
> ---
>
> ### **References**
>
> [1] Yang, Y., and Kuchibhotla, A. K. (2024). Selection and aggregation of conformal prediction sets. _Journal of the American Statistical Association_, 1–13.
>
> [2] Liang, R., Zhu, W., and Barber, R. F. (2024). Conformal prediction after efficiency-oriented model selection. _arXiv preprint arXiv:2408.07066_.
>
> [3] Gauraha, N., and Spjuth, O. (2021). Synergy conformal prediction. _Conformal and Probabilistic Prediction and Applications. PMLR_.

---

### Decision · Program_Chairs · 2025-05-01

**Decision:**

Accept (poster)

**Comment:**

In this paper, the authors propose an ensemble approach to conformal prediction, which is based on the idea of averaging multiple conformity score functions. The weights of the individual scores (in the weighted average) are optimised so as to maximise efficiency (i.e., minimise expected prediction set size) while keeping the validity guarantee. A theoretical analysis establishes finite-sample coverage guarantees. Moreover, empirical studies on benchmark data show that weighted averaging systematically outperforms single-score methods.

Overall, this is a convincing paper with a solid contribution to conformal prediction, although the idea as such is very simple. The reviewers had some comments and made some minor suggestions for improvement, which could be addressed by the authors in their rebuttal.